# The Shape of Addition: Geometric Structures of Arithmetic in Large Language Models

**Liuyuan Wen** [† 1]  **Xun Zhu** [† 1]  **Lihao Huang** [† 1]  **Wenbin Li** [* 1]  **Yang Gao** [1]

## Abstract

Large Language Models exhibit paradoxical fragility in fundamental arithmetic, implying a disconnect between internal computation and discrete output. By analyzing the residual stream geometry during multi-operand addition, we identify the Iso-Raw-Sum Trajectory (IRST), a geometric structure where representations are anchored by semantic digits and modulated by continuous carry fibers. We propose the Noisy Quantization Model to explain this geometry, framing arithmetic errors as Geometric Slippages caused by internal neural noise pushing a continuous, latent Carry Potential across quantization thresholds. This geometric framework further elucidates Probe Versatility, explaining how lightweight probes can disentangle coexisting latent signals (such as ground truth versus hallucination) from a single activation vector. Finally, we validate these insights through a geometric consistency check method that effectively detects and corrects these quantization failures during inference. Our code is available at https://github.com/RL-MIND/Shape-of-Addition.

## 1. Introduction

Large Language Models (LLMs) have demonstrated remarkable capabilities in complex mathematical reasoning, achieving high performance on benchmarks such as GSM8K (Cobbe et al., 2021) and MATH (Hendrycks et al., 2021). However, a paradox remains: despite their proficiency in high-level problem solving, LLMs exhibit surprising fragility in fundamental algorithmic primitives (Li et al., 2025), particularly multi-digit arithmetic (Nanda et al.,

[†]Equal contribution [*]Corresponding author [1]State Key Laboratory of Novel Software Technology, Nanjing University, Nanjing 210023, China. Correspondence to: Wenbin Li <liwenbin@nju.edu.cn>.

*Proceedings of the 43rd International Conference on Machine Learning*, Seoul, South Korea. PMLR 306, 2026. Copyright 2026 by the author(s).

2023). Even sophisticated models frequently commit "off-by-one" errors as the number of operands increases, suggesting a fundamental disconnect between their internal computation and final discrete output. While prior research has attempted to model these behaviors through either symbolic manipulation (Quirke et al., 2025) or geometric representations (Nanda et al., 2023; Kantamneni & Tegmark, 2025), the precise mechanism linking internal activation geometries to specific output failures remains elusive.

Crucially, recent probing studies have deepened this mystery (Yan, 2025; Baeumel et al., 2025b). Lightweight probes can now accurately detect error signals and even decode the ground truth from the residual stream of a failing model (Su et al., 2024; Sun et al., 2025). This implies a critical disconnect: the model internally encodes the correct information, yet fails to output it. While existing probes serve as effective diagnostic tools, they identify *when* a model errs without explaining *how* representation geometry mechanistically induces specific failures like carry propagation errors.

In this work, we investigate the fine-grained internal representations of LLMs performing a challenging arithmetic task: the addition of three (or more) 10-digit integers. Unlike previous studies that focus on the first token (Sun et al., 2025), we analyze the residual stream activations at every generated digit position (as shown in Figure 1). Our probing experiments reveal a phenomenon we term *probe versatility*. We find that lightweight probes (e.g., logistic regression, MLPs) can simultaneously decode diverse signals—including the Ground Truth, Model Output, Input Carry and so on—from a single activation vector. The coexistence of these signals implies that the representation space is actually a highly structured manifold. This raises a fundamental representational question: *What is the geometric structure underlying LLM arithmetic representations that allows ground truth and hallucination to coexist?*

To decipher this structure, we employ UMAP (McInnes et al., 2018) as dimensionality reduction to visualize the internal activation manifolds, primarily at the final layer (Figure 2). We discover that representations are organized into **Iso-Raw-Sum Trajectories (IRSTs)**—continuous fibers of constant raw sum that thread through a hierarchical geom-

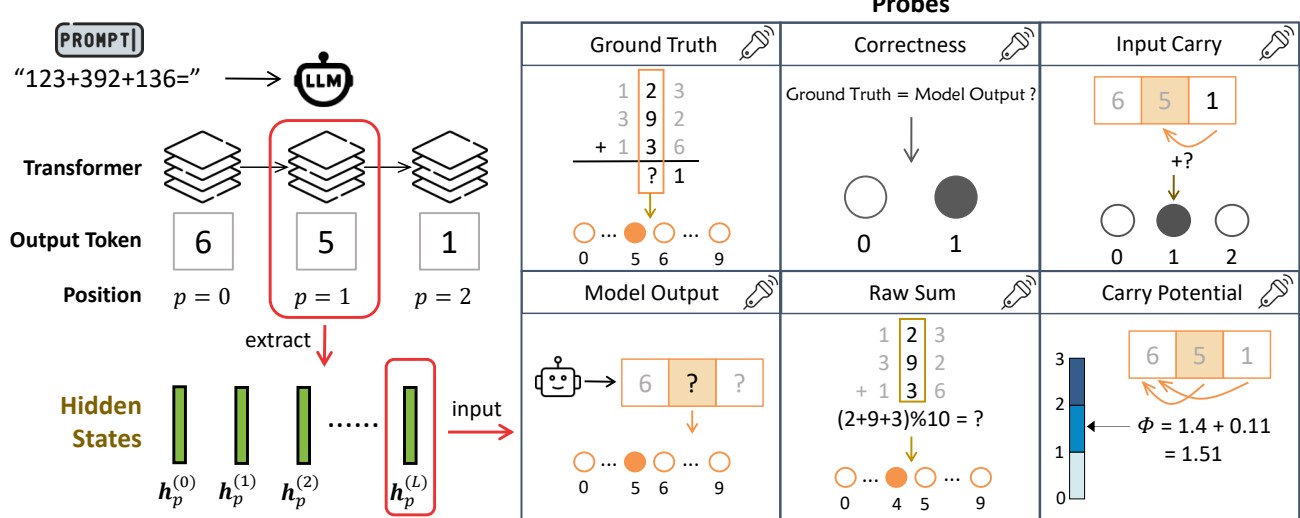

Figure 1. **Overview of our probing framework. (Left)** The LLM performs multi-operand addition (e.g., $123 + 392 + 136$) in an autoregressive manner. At each generation step (e.g., $p = 1$, corresponding to the tens digit), we extract the hidden state vectors $h_p^{(l)}$ (mainly focusing on the final layer $L$). **(Right)** We train versatile probes on these activation vectors to decode several critical arithmetic variables, including discrete states (Ground Truth, Model Output, Correctness, Input Carry, Raw Sum) and continuous variables (Carry Potential $\Phi$, defined in Section 5). The ability to decode these diverse and interrelated signals from a single vector motivates our geometric analysis of the residual stream.

etry composed of a macroscopic backbone of digit basins and a microscopic texture of input carry states. Within this framework, arithmetic errors are characterizable as *geometric slippages*—continuous drifts along an IRST that push the representation across the decision boundaries of adjacent digit basins due to ambiguity in carry representation.

Building on these observations, we propose the **Noisy Quantization Model**, positing that LLMs estimate a continuous *Carry Potential* which is subsequently discretized. This framework characterizes arithmetic failures as noise-induced threshold crossings, accurately predicting the periodic *bathtub* error distribution observed in our experiments (Figure 5). Furthermore, this geometric perspective helps us demonstrate that the *probe versatility* phenomenon is a direct function of the manifold's topological separability rather than mere information extractability.

Finally, we validate this framework through an inference-time self-correction method via dual-stream consistency. By enforcing logical alignment between decoded local (Raw Sum) and global (Carry Potential) signals, we significantly recover performance. This confirms that the model's internal representations retain the correct mathematical components even when the final token selection is erroneous.

In summary, the main contributions of this work are:

- We identify the **Iso-Raw-Sum Trajectory (IRST)**, revealing how LLMs represent arithmetic states through a hierarchy of raw sum fibers and digit basins.

- We propose the **Noisy Quantization Model**, which mechanistically models arithmetic errors as quantization failures of a continuous *Carry Potential*.

- We provide a geometric elucidation of probing phenomenon, showing that probing performance hierarchy is structurally determined by the manifold's topology.

- We introduce a dual-stream consistency check method for inference-time intervention, validating the retention of correct latent signals even in erroneous cases.

## 2. Preliminaries

We investigate the internal mechanisms of LLMs during multi-operand addition. This section establishes the mathematical definitions and the probing framework used to infer latent internal states, as illustrated in Figure 1.

### 2.1. Mathematical Framework

We consider the task of adding $n$ integers, denoted as $A_0, A_1, \ldots, A_{n-1}$. Each integer $A_i$ consists of $m$ decimal digits. Let $a_{i,k} \in \{0, 1, \ldots, 9\}$ denote the digit of $A_i$ at the $10^k$ position (where the units place corresponds to $k = 0$). Therefore, each addend can be expressed as $A_i = \sum_{k=0}^{m-1} a_{i,k} \cdot 10^k$. Let $S$ be the sum of these integers, i.e., $S = \sum_{i=0}^{n-1} A_i$.

We construct a dataset $\mathcal{D}$ of $N$ addition problems. The LLM generates the sum token-by-token in an autoregressive

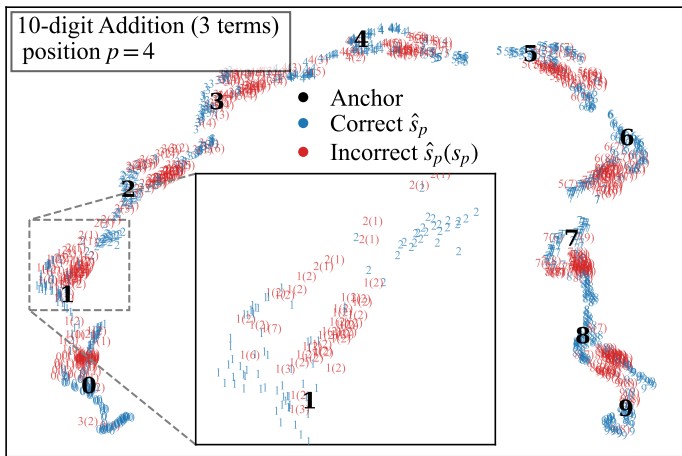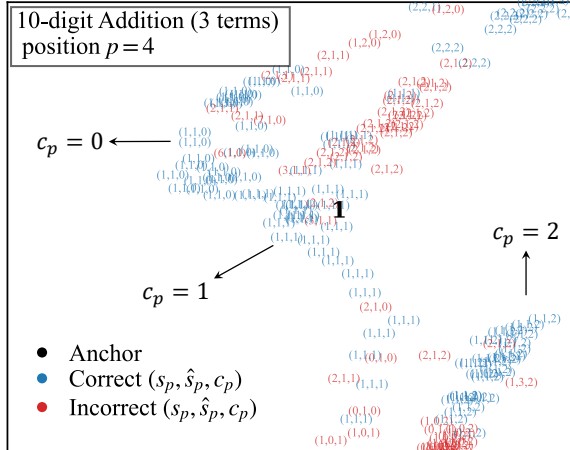

*Figure 2.* **2D UMAP visualization of the arithmetic manifold. (Left) Macroscopic Backbone:** Global geometry of $h_p^{(L)}$ ($p = 4$) organized around digit Anchors (0–9). Blue points denote correct samples labeled as $\hat{s}_p$; red points denote errors labeled as $\hat{s}_p(s_p)$. The inset highlights high-error transition zones between digit basins. **(Right) Microscopic Texture:** Magnified view around Anchor 1 labeled with $(s_p, \hat{s}_p, c_p)$, showing that representations within a digit basin are further stratified into distinct fibers by the input carry state $c_p$. (The geometric consistency in high-dimensional space is validated in Appendix C.)

manner from the most significant digit to the least. We define the generation step as the *position p*. If the target sum $S$ has a length of $P$ digits, the generation position $p \in \{0, \ldots, P-1\}$ corresponds to the mathematical significance $k = P - 1 - p$.

Crucially, the arithmetic state at any position $p$ is governed by three fundamental variables: the Raw Sum $r_p = \sum_{i=0}^{n-1} a_{i,p}$, which is the column-wise sum of input digits; the Input Carry $c_p$, propagated from the lower-order position; and the Ground Truth Digit $s_p$. These satisfy the modular relation $s_p \equiv (r_p + c_p) \pmod{10}$.

To characterize the model's internal state, we introduce parallel variables for the model's behavior. Let $\hat{s}_p$ denote the Predicted Digit generated by the model. We further postulate two latent variables: the Predicted Raw Sum $\hat{r}_p$ and the Predicted Input Carry $\hat{c}_p$, representing the internal information the model utilizes to generate $\hat{s}_p$. We aggregate these into a state tuple for each position:

$$\mathcal{X}_p = (s_p, \hat{s}_p, c_p, \hat{c}_p, r_p, \hat{r}_p). \tag{1}$$

This formalism allows us to distinguish between the mathematical truths $(s_p, c_p, r_p)$ and the model's internal representations $(\hat{s}_p, \hat{c}_p, \hat{r}_p)$.

### 2.2. Internal Activations and Probing

For every problem in the dataset $\mathcal{D}$ and at each generation position $p$, we extract the internal hidden states $h_p^{(l)}$ from layer $l$ (primarily focusing on the final layer $L$, layer-wise analysis is provided in Appendix K). To decode the information encoded in these representations, we train lightweight probes on $h_p^{(L)}$. We target the discrete variables defined in the previous section: the ground truth digit $s_p$, the predicted

digit $\hat{s}_p$, the input carry $c_p$, and the raw sum $r_p$. Additionally, we investigate continuous variables such as the *Carry Potential* $\Phi$, a scalar value modeling the accumulation of arithmetic value from the context, which is defined in Section 5.

### 2.3. Latent Carry States

While $s_p$, $\hat{s}_p$, $c_p$, and $r_p$ are directly observable, the internally predicted $\hat{c}_p$ and $\hat{r}_p$ are latent and must be inferred. We assume that the model generally performs the local addition operation correctly (i.e., $\hat{r}_p \approx r_p$), see Appendix D for a validation. So that errors in $\hat{s}_p$ primarily stem from incorrect carry states. By analyzing the modular arithmetic relations, we can map the observed output errors to internal carry deviations. Specifically, if the model underestimates the digit by 1, it implies a leakage in the internal carry ($\hat{c}_p = c_p - 1$); conversely, if it overestimates by 1, it implies a hallucinated carry ($\hat{c}_p = c_p + 1$). A detailed derivation is provided in Appendix E.

## 3. Representational Geometry

### 3.1. Experimental Setup

To empirically investigate the latent structure of arithmetic reasoning, we conduct experiments using the Qwen3-4B model (Yang et al., 2025), which consists of $L = 36$ transformer layers. We construct a dataset $\mathcal{D}$ containing $N = 10,000$ addition problems, each involving three 10-digit integers. In this 3-term addition setup, the input carry $c_p$ at any position can take values in $\{0, 1, 2\}$. We focus our analysis on the hidden states $h_p^{(L)}$ extracted from the final

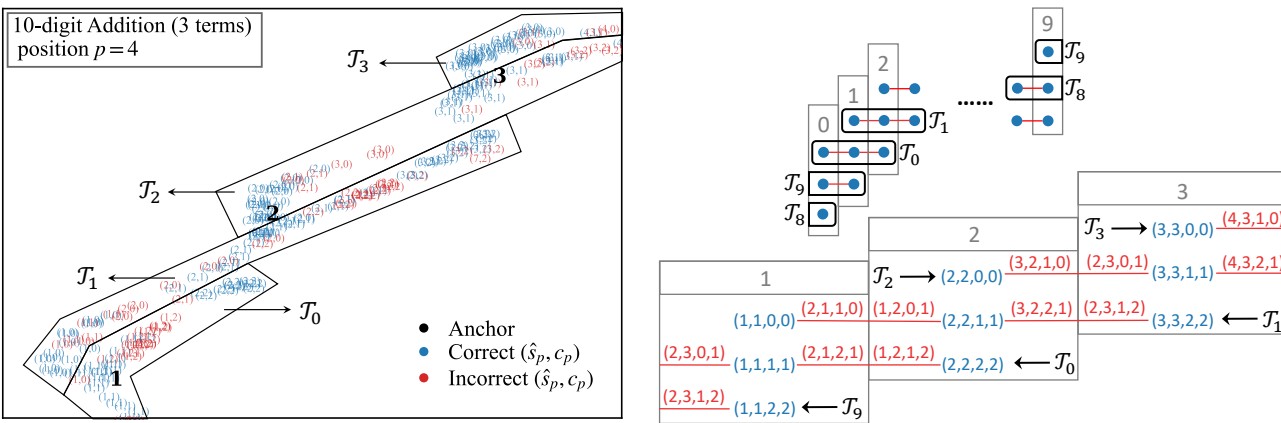

*Figure 3.* **The Iso-Raw-Sum Trajectory (IRST) framework of the arithmetic manifold. (Left)** Magnified UMAP projection around digit Anchors 1, 2, and 3. Points are labeled with $(\hat{s}_p, c_p)$. The geometry reveals distinct IRSTs ($\mathcal{T}_0 \sim \mathcal{T}_3$) that act as continuous "threads" piercing through adjacent digit basins. For instance, $\mathcal{T}_1$ (where $r_p \bmod 10 = 1$) connects stable nodes $(1,0) \leftrightarrow (2,1) \leftrightarrow (3,2)$ as the input carry increases. **(Top-Right)** Abstract representation of the global manifold, illustrating how parallel raw-sum fibers ($\mathcal{T}_r$) organize the representation space across all digit basins (0–9). **(Bottom-Right)** Detailed state-level schematic of the IRST framework. Gray boxes represent digit basins; states are defined by the tuple $(s_p, \hat{s}_p, c_p, \hat{c}_p)$. Blue denotes correct predictions, while red denotes *geometric slippages* (errors). These errors are concentrated at transition zones between basins along an IRST, resulting from the model's miscalculation of the input carry ($\hat{c}_p = c_p \pm 1$).

layer at position $p = 4$[1]. Analysis of other interior positions yields qualitatively similar findings, while boundary positions are discussed in Appendix G.

We visualize the high-dimensional activation space using UMAP (McInnes et al., 2018) dimensionality reduction with *cosine* distance as the distance metric (see Appendix B for other reduction methods). To provide a semantic reference, we include the output unembedding vectors of the digit tokens 0 through 9 in the reduction process, which serve as fixed *Anchors*[2] in the resulting 2D map. The visualization, presented in Figure 2, reveals a highly organized geometric manifold that encodes the model's arithmetic state through a hierarchical geometric structure. More detailed experimental settings are provided in Appendix A.

### 3.2. Geometric Structure

The geometry of the last-layer activations exhibits a clear separation between global semantic identity and local arithmetic state. We characterize this structure as having a macroscopic *backbone* modulated by a microscopic *texture*. The macroscopic backbone is dominated by the identity of the output tokens. As seen in the left panel of Figure 2, the activation vectors are strongly clustered into ten distinct basins, each centered around a specific digit anchor. This indicates that the primary organization of the residual stream at the final layer is determined by the categorical selection of the next token.

Within each digit basin, the representation space possesses a fine-grained microscopic texture. As illustrated in the magnified view around Anchor 1 in the right panel of Figure 2, the data points are not distributed uniformly. Instead, they are stratified into parallel sub-manifolds or *fibers* determined by the ground truth input carry $c_p$. Specifically, for any digit basin, we observe three distinct clusters corresponding to the states $c_p \in \{0, 1, 2\}$. This suggests that the model internalizes the underlying algebraic structure (Chang et al., 2024) to distinguish the source of a digit. For instance, differentiating "a 3 derived from no carry" from "a 3 derived from a carry of 1".

The distribution of correct (blue) and incorrect (red) samples further elucidates the reliability of this manifold in Figure 2. Correct predictions cluster densely at the center of these carry-specific fibers, representing stable states of computation. Conversely, errors predominantly occur in the sparse transition zones between digit basins or between carry fibers. As shown in the inset of the left panel, incorrect samples often lie on the fringes of their ground truth digit basins, physically drifting toward adjacent anchors. This drift manifests empirically as the prevalent "off-by-one" errors—a common failure mode cataloged in mathematical reasoning benchmarks (Zhang & Graf, 2025), confirming that geometric adjacency on the manifold aligns with numerical adjacency. This spatial distribution suggests that arithmetic failures are essentially geometric instabilities, where *the internal representation fails to align with the correct geometric trajectory.*

---

[1]We select $p = 4$ as a representative middle position.

[2]These anchors serve exclusively as auxiliary landmarks for semantic grounding; their inclusion does not alter the intrinsic geometry of the activation vectors in the UMAP projection.

# 4. Geometric Analysis of Internal States

Building on the UMAP visualization of the last-layer activations, we propose a geometric framework called the **Iso-Raw-Sum Trajectory (IRST)** to interpret how the LLM encodes arithmetic states (Figure 3).

## 4.1. Iso-Raw-Sum Trajectories (IRST)

The key insight connecting the discrete digit clusters is the conservation of the *raw sum* $r_p$. We define an **IRST**, denoted as $\mathcal{T}_r$, as a continuous manifold connecting all internal states that share the same raw sum $r_p = r$. Recalling the identity $\hat{s}_p = (r_p + \hat{c}_p) \pmod{10}$, a trajectory $\mathcal{T}_r$ does not stay within a single digit basin but traverses across adjacent basins as the carry $\hat{c}_p$ varies.

As illustrated in Figure 3 (Left), consider the trajectory $\mathcal{T}_1$ (where raw sum $r_p = 1$). It acts as a continuous *thread* that pierces through the boundaries of Anchors 1, 2, and 3, connecting three stable fixed nodes in a linear sequence:

$$\mathcal{T}_1 : \underbrace{(1,1,0,0)}_{\text{Anchor 1, } c=0} \leftrightarrow \underbrace{(2,2,1,1)}_{\text{Anchor 2, } c=1} \leftrightarrow \underbrace{(3,3,2,2)}_{\text{Anchor 3, } c=2} . \quad (2)$$

Here, we use the tuple notation $(s_p, \hat{s}_p, c_p, \hat{c}_p)$. The model transitions between output digits $s$ and $s+1$ by sliding along these fixed raw sum fibers, driven by the shift in the internal carry state.

## 4.2. Geometric Slippage on Trajectories

In this framework, arithmetic errors are not random noise but specific *Geometric Slippages* along an IRST. They occur when activation vectors drift onto unstable segments connecting stable nodes, rather than aligning with the stable nodes themselves.

Focusing on the transition between Anchor 1 and Anchor 2 (see Figure 3, Bottom-Right), we identify two distinct error paths corresponding to different raw sums:

**Scenario 1: Slippage on $\mathcal{T}_1$ (Raw Sum = 1).** This trajectory bridges the stable node $(1,1,0,0)$ and $(2,2,1,1)$. Errors found on the segment connecting these nodes represent a specific confusion between carry 0 and carry 1. In the case of *Overestimation (Hallucination)*, characterized by state $(1,2,0,1)$, the ground truth requires $c = 0$, but the model hallucinates $\hat{c} = 1$, geometrically pushing the representation from Anchor 1 toward Anchor 2. Conversely, *Underestimation (Leakage)* corresponds to state $(2,1,1,0)$; here, the ground truth involves $c = 1$, but the model misses the carry ($\hat{c} = 0$), causing the vector to slide back from the correct basin of Anchor 2 into Anchor 1.

**Scenario 2: Slippage on $\mathcal{T}_0$ (Raw Sum = 0).** Similarly, $\mathcal{T}_0$ bridges $(1,1,1,1)$ and $(2,2,2,2)$, involving confusion

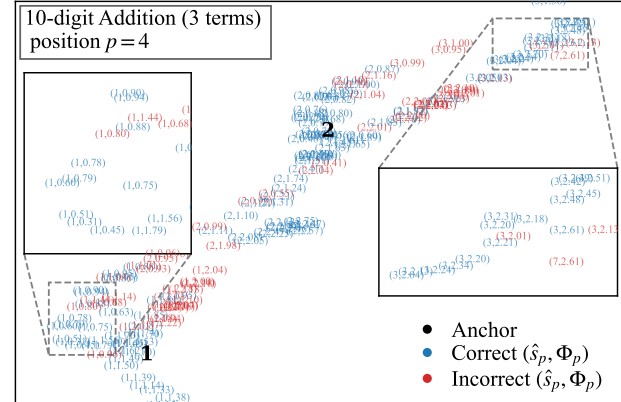

*Figure 4.* **Geometric unfolding of the IRST by Carry Potential.** Visualization of the trajectory $\mathcal{T}_1$, where each point is labeled with the tuple $(\hat{s}_p, \Phi_p)$. **(Insets)** The geometry correlates strongly with the continuous value of the theoretical carry potential $\Phi_p$. The left inset highlights the region where $\Phi_p < 1$ (approaching the threshold from below), corresponding to the stable node of $c_p = 0$. The right inset displays the region where $\Phi_p > 2$, corresponding to the node of $c_p = 2$. The smooth spatial progression of $\Phi_p$ values along the manifold confirms that the IRST is physically organized by the continuous *thrust* to carry, validating the continuity assumption of the Noisy Quantization Model.

between carry 1 and 2. Here, hallucination manifests as $(1,2,1,2)$, where the model erroneously jumps to $\hat{c} = 2$ despite $c = 1$. Leakage appears as $(2,1,2,1)$, where the model fails to sustain the carry, retreating to the lower state $\hat{c} = 1$.

The geometric nature of arithmetic errors is therefore defined by the proximity to the decision boundary between anchors along a specific IRST. Correct predictions cluster densely around the stable nodes, while errors are distributed along the sparse connecting paths. In these transitional regions, the unembedding layer struggles to distinguish between the two digits, as the activation contains mixed signals: the raw sum suggests continuity, but the ambiguous carry state places the vector in a "no-man's-land" between two semantic truths.

# 5. Formal Modeling of Carry Dynamics

Our geometric analysis reveals that the IRSTs act as continuous manifolds connecting discrete digit anchors. This continuity suggests that the LLM's internal representation of arithmetic is not purely symbolic. Instead, we hypothesize that the model maintains a continuous, latent variable representing the "pressure" to carry, which is subsequently discretized. In this section, we verify this hypothesis by proposing the **Noisy Quantization Model**. We demonstrate that arithmetic errors are essentially predictable consequences of signal detection noise near quantization boundaries.

## 5.1. Carry Potential ($\Phi$)

We define the *Carry Potential*, denoted as $\Phi_p \in \mathbb{R}_{\geq 0}$, as the theoretical accumulation of numerical value flowing from lower-order positions (the right context) into the current position $p$. Unlike the discrete ground truth carry $c_p \in \mathbb{Z}$, the potential $\Phi_p$ is a continuous scalar derived from the weighted sum of raw sums in the context:

$$\Phi_p = \sum_{j=1}^{P-1-p} \frac{r_{p+j}}{10^j}, \qquad (3)$$

where $r_{p+j}$ is the raw sum of the digits at relative position $j$ to the right of $p$. Physically, $\Phi_p$ represents the *thrust* or *momentum* of the carry signal, and the discrete ground truth carry corresponds to its integer floor: $c_p = \lfloor \Phi_p \rfloor$. For example, as illustrated in Figure 1, the rightward context yields a potential of $\Phi = 1.4 + 0.11 = 1.51$. This continuous value physically represents the *thrust* to carry, which is subsequently quantized to the discrete carry $c_p = \lfloor 1.51 \rfloor = 1$.

This theoretical definition is strongly corroborated by our empirical observations. As shown in Figure 4, when we label the activation manifold with the calculated values of $\Phi_p$, we observe a smooth spatial gradient along the IRST. The representation explicitly encodes the continuous magnitude of $\Phi_p$, placing states with $\Phi_p \approx 0.9$ (high risk of carry) spatially closer to the next carry basin than states with $\Phi_p \approx 0.1$.

## 5.2. Noisy Quantization Hypothesis

We posit that the LLM estimates this external potential $\Phi_p$ via its internal representations. We further hypothesize that, due in part to the dilution of attention over long contexts and the limited precision of superposition, this estimation is corrupted by neural noise. We model the *Perceived Carry Potential*, $\hat{\Phi}_p$, as:

$$\hat{\Phi}_p = \Phi_p + \epsilon, \quad \epsilon \sim \mathcal{N}(0, \sigma^2), \qquad (4)$$

where $\epsilon$ represents additive Gaussian noise with zero mean and variance $\sigma^2$. The parameter $\sigma$, which we term the *Cognitive Noise Level*, encapsulates the effective uncertainty in the model's estimation imposed by the current task complexity. The model then generates the discrete predicted input carry $\hat{c}_p$ by performing a quantization operation (flooring) on this noisy signal:

$$\hat{c}_p = \lfloor \hat{\Phi}_p \rfloor = \lfloor \Phi_p + \epsilon \rfloor. \qquad (5)$$

Errors occur when the noise $\epsilon$ is sufficient to push the perceived potential $\hat{\Phi}_p$ across an integer quantization boundary.

## 5.3. Bathtub Error Rate

This formulation constitutes a threshold detection problem. The likelihood of an error depends on the distance of the

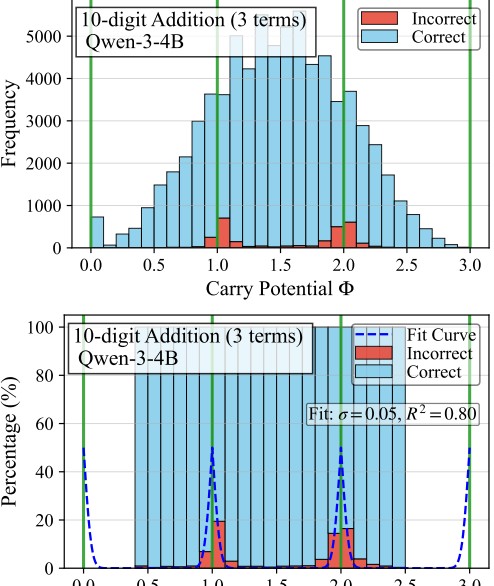

*Figure 5.* **Empirical validation of the Noisy Quantization Model. (Top)** The distribution of Carry Potential $\Phi$ across all generated positions $p$ in the dataset. Green vertical lines indicate integer quantization thresholds $(1.0, 2.0, \ldots)$. **(Bottom)** The conditional error rate as a function of $\Phi$. The empirical data (red bars, off-by-one errors only) exhibits a distinct periodic *bathtub* shape, spiking near integer boundaries (intervals with low sample density are omitted). The theoretical error curve (dashed blue line), derived from Equation (6), fits the data with high fidelity ($R^2 = 0.80$), estimating the model's internal noise level at $\sigma \approx 0.05$.

potential $\Phi_p$ to the nearest integer boundary.

Let $\delta(\Phi) = \Phi \pmod 1$ be the fractional part of the potential, the total probability of an off-by-one carry error given a potential $\Phi$ is the sum of the probabilities of *Underestimation (Leakage)* and *Overestimation (Hallucination)*:

$$P(\text{Error} \mid \Phi) = \underbrace{Q\left(\frac{\delta(\Phi)}{\sigma}\right)}_{\text{Leakage Risk}} + \underbrace{Q\left(\frac{1 - \delta(\Phi)}{\sigma}\right)}_{\text{Hallucination Risk}}, \quad (6)$$

where $Q(\cdot)$ is the standard Q-function. Detailed derivation provided in Appendix F. This equation predicts a periodic *bathtub* error distribution: error rates spike when $\Phi$ is close to an integer $i$ (*metastability*) and vanish when $\Phi$ is near $i + 0.5$ (*robust plateaus*).

To empirically validate this prediction, we computed the analytical carry potential $\Phi$ across the entire dataset $\mathcal{D}$. Both the global distribution of potentials (Figure 5, Top) and the conditional error rate (Figure 5, Bottom) reveals a striking periodic *bathtub* pattern. Error rates spike drastically near integer boundaries (e.g., $\Phi \approx 1.0, 2.0$), identifying them as regions of *metastability* where the signal is most ambiguous. Conversely, the error rates reach their minima at half-integer values (e.g., $\Phi \approx 1.5$). These *robust plateaus* represent

*Table 1.* Probing accuracy on the final layer activations ($p = 4$). Datasets are individually balanced for each probe target.

| TARGET VARIABLE | SYMBOL | ACC |
|---|---|---|
| GROUND TRUTH | $s_p$ | 94.85% |
| MODEL OUTPUT | $\hat{s}_p$ | 98.81% |
| CORRECTNESS | $\mathbb{I}(\hat{s}_p \neq s_p)$ | 82.41% |
| RAW SUM (MOD 10) | $r_p \pmod{10}$ | 98.60% |
| INPUT CARRY | $c_p$ | 96.84% |
| CARRY POTENTIAL | $\Phi_p$ | 92.08%(FLOORED) |

the states where the signal is maximally distinct from the quantization thresholds.

By fitting the theoretical curve to this empirical distribution, we extract a noise parameter of $\sigma \approx 0.05$. This value quantifies the model's processing uncertainty, reflecting the precision limit of internal signal processing specific to this computational context. The close alignment between theory and observation confirms that arithmetic errors are mechanistically determined by this internal noise pushing the carry potential across quantization boundaries. More investigation into how this noise scales with task complexity is provided in Appendix G.

## 6. Trajectory-Level Validation and Causal Steering

The IRST geometry illustrated in Figures 3 and 4 predicts more than a visually plausible 2D embedding: along any IRST, arithmetic states should be ordered by a continuous carry coordinate, and moving a representation along that coordinate should causally change the generated digit. To test this directly, we define a trajectory-local steering direction between two adjacent stable centroids $\boldsymbol{\mu}_a$ and $\boldsymbol{\mu}_b$ on the same IRST:

$$\vec{v}_{steer} = \frac{\boldsymbol{\mu}_b - \boldsymbol{\mu}_a}{\|\boldsymbol{\mu}_b - \boldsymbol{\mu}_a\|}. \tag{7}$$

We then intervene on a final-layer pre-norm activation $\boldsymbol{h}$ by

$$\tilde{\boldsymbol{h}}(\alpha) = \boldsymbol{h} + \alpha \cdot \vec{v}_{steer}, \tag{8}$$

where $\alpha > 0$ moves the state toward the higher-carry basin and $\alpha < 0$ suppresses the carry signal. If the carry potential is genuinely represented along this direction, samples near a quantization boundary should flip under smaller perturbations than deep in-basin states.

We evaluate this prediction on a representative trajectory $\mathcal{T}_3$ at $p = 4$, where the adjacent stable states $(3, 3, 0)$, $(4, 4, 1)$, and $(5, 5, 2)$ are all well populated. For this case, we instantiate the steering vector using $\boldsymbol{\mu}_a = \boldsymbol{\mu}_{(4,4,1)}$ and $\boldsymbol{\mu}_b = \boldsymbol{\mu}_{(5,5,2)}$. In Figure 6 (Left), plotting the analytical Carry Potential $\Phi$ against the cosine distance to $\boldsymbol{\mu}_{(4,4,1)}$ yields the predicted V-shaped progression, with off-by-one

errors such as $(4, 5, 1)$ and $(5, 4, 2)$ concentrated in the transition regions rather than forming a separate cluster. In Figure 6 (Right), the boundary-adjacent states $(4, 5, 1)$ and $(5, 4, 2)$ switch at substantially smaller perturbation magnitudes ($\alpha \approx -0.1$ and $\alpha \approx 0.3$, respectively) than the stable states $(5, 5, 2)$ and $(4, 4, 1)$ ($\alpha \approx -0.5$ and $\alpha \approx 0.5$). This separation of critical thresholds supports the claim that the model's arithmetic decision is governed by a carry-like continuous direction. Additional analysis are deferred to Appendix C.

## 7. Geometric Origins of Probing Phenomenon

Our discovery offers a unified geometric lens to interpret the varying performance of linear probes reported in literature. We trained same logistic regression probes on a same balanced datasets at $p = 4$. As shown in Table 1, the accuracy differences are essentially direct measures of the geometric separability of the underlying manifold.

**Model Output vs. Ground Truth.** The probe for Model Output achieves near-perfect accuracy (98.81%), effectively mimicking the unembedding matrix to identify the Voronoi basin of the predicted anchor. In contrast, the Ground Truth probe lags significantly (94.85%). This gap confirms our *geometric slippage* hypothesis: Activation vectors physically drift to incorrect basins during errors. For the probe to recover the ground truth in these cases, it would need to map the incorrect basin back to the correct label, which contradicts the local geometry.

**Correctness as Boundary Detection.** The Correctness probe yields the lowest accuracy (82.41%). Geometrically, this probe acts as a stability detector, attempting to distinguish samples deep within a basin (correct) from those in the transition zones (likely errors). The moderate performance implies that the manifold is continuous; there is no sharp error flag or discrete boundary separating correct and incorrect states, only a smooth gradient of ambiguity along the IRST.

**Decoding the IRST Structure.** The high accuracy of the Raw Sum probe (98.60%) supports the existence of distinct IRSTs. Since most arithmetic errors are off-by-one carry shifts, the representation slides along the same IRST rather than jumping to a different one, preserving the linear separability of the raw sum. Similarly, the Input Carry probe (96.84%) successfully disentangles the parallel fibers (carry 0, 1, 2) within these trajectories.

**Continuous Potential vs. Discrete Carry.** The Carry Potential probe, trained via regression, achieves 92.08% accuracy (after quantization). This slightly lower performance compared to the discrete carry probe reflects the complex-

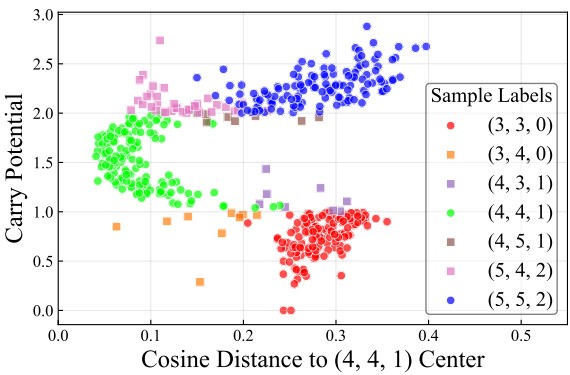 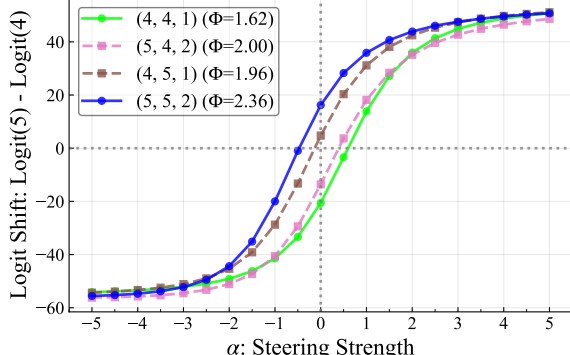

*Figure 6.* **Representative trajectory-level validation on $\mathcal{T}_3$.** The markers are labeled as $(s_p, \hat{s}_p, c_p)$. Circles denote correct predictions and squares denote errors. **(Left)** Empirical projection of last-layer activations for samples in $\mathcal{T}_3$. The x-axis shows the cosine distance from the central centroid $(4, 4, 1)$. The manifold exhibits a clear V-shaped progression connecting stable basins. Crucially, error states such as $(4, 3, 1)$ and $(5, 4, 2)$ cluster in the transition regions between stable fibers, validating the *geometric slippage* hypothesis. **(Right)** Causal steering results. We inject the carry vector $\vec{v}_{steer}$ (derived from contrastive means) and measure the Logit Shift (Logit(5) − Logit(4)). The results show characteristic sigmoid phase transitions: stable states require larger $|\alpha|$ to flip than metastable error states near the boundary.

ity of mapping the continuous position along an IRST to a discrete integer. The probe is essentially measuring the distance traveled along the trajectory; the fact that it correlates highly with the discrete carry validates that the IRST is physically organized by this continuous magnitude.

# 8. Inference-Time Self-Correction

Building on the Noisy Quantization model, we propose a geometric inference-time intervention acting as a causal probe to validate the IRST structure. By mitigating quantization noise through logical consistency constraints, we show that the underlying arithmetic signal remains intact and recoverable.

## 8.1. Dual-Stream Consistency Protocol

We interpret the activation vector $\boldsymbol{h}_p^{(L)}$ at the final layer through two orthogonal lightweight probes. First, we decode the *Local Calculation Stream* using a classification probe $f_{\theta_r}$ to obtain the predicted raw sum digit $\hat{r}_p$. Second, we decode the *Global Context Stream* using a regression probe $f_{\theta_\phi}$ to estimate the continuous carry potential $\hat{\Phi}_p$. As demonstrated in Section 5, $\hat{\Phi}_p$ acts as a proxy for the cumulative context information, while $\hat{r}_p$ represents the immediate column-wise arithmetic.

In a consistent arithmetic system, the generated token $\hat{s}_p$ must satisfy the modular identity $\hat{s}_p \equiv (\hat{r}_p + c) \pmod{10}$ for some valid carry $c$. However, due to the inherent cognitive noise $\sigma$ in estimating $\hat{\Phi}_p$, enforcing a strict integer carry $c = \lfloor \hat{\Phi}_p \rfloor$ is prone to false positives near quantization boundaries. To address this, we introduce a robust consistency check. We define a set of *Plausible Carries* $\mathcal{K}_p(\delta)$ derived from the $\delta$-neighborhood of the estimated potential:

*Table 2.* Performance comparison of different correction methods on Token-level Accuracy, True Positive Correction, and False Positive Preservation.

| METHOD | TOKEN ACC | TP CORR | FP PRES |
|---|---|---|---|
| ORIGINAL | 86.26% | / | / |
| RE-PROMPTING | 79.90% | 0.08% | **99.98%** |
| STEERING | 88.27% | 30.58% | 96.97% |
| REPLACEMENT | 89.13% | 31.73% | 97.65% |
| OURS($\delta = 0$) | 87.27% | **44.39%** | 94.07% |
| OURS ($\delta = 0.1$) | **89.56%** | 30.46% | 98.13% |

$$\mathcal{K}_p(\delta) = \left\{ \lfloor \phi \rfloor \mid \phi \in [\hat{\Phi}_p - \delta, \hat{\Phi}_p + \delta] \right\}. \qquad (9)$$

The generation $\hat{s}_p$ is deemed consistent if it can be explained by the local raw sum $\hat{r}_p$ and any plausible carry $c \in \mathcal{K}_p(\delta)$.

If the model's output $\hat{s}_p$ fails this consistency check, it indicates a geometric divergence where the representation has drifted off the valid IRST. In such cases, we intervene by overriding the output logits: $\hat{s}_{new} = (\hat{r}_p + \lfloor \hat{\Phi}_p \rfloor) \pmod{10}$. This effectively projects the divergent state back onto the correct manifold fiber.

## 8.2. Experimental Results

We evaluate our method against baselines, including re-prompting (Sun et al., 2025), linear steering (Bhalla et al., 2024) and hard replacement (using a Ground Truth probe). As shown in Table 2, our method ($\delta = 0.1$) achieves the highest accuracy (89.56%). Crucially, these results serve as a causal validation of the IRST geometry rather than a mere performance boost. The ability to correct errors using only internal consistency confirms that the model retains correct latent information despite erroneous output quantization.

*Table 3.* Ablation study disentangling raw-sum and carry information for the dual-stream correction mechanism ($\delta = 0$, Qwen3-4B, 3-term addition).

| METHOD | TOKEN ACC | TP CORR | FP PRES |
|--------|-----------|---------|---------|
| R+C | 86.7% | 42.3% | 93.9% |
| R+TC | 96.0% | 69.3% | 99.0% |
| TR+C | 90.5% | 65.4% | 94.6% |

R+C denotes *Raw-Sum Probe + Carry Probe*, R+TC denotes *Raw-Sum Probe + True Carry*, and TR+C denotes *True Raw-Sum + Carry Probe*.

Moreover, the sensitivity to $\delta$ corroborates the Noisy Quantization Model. While $\delta = 0$ yields the highest correction rate (TP Corr: 44.39%), it aggressively overwrites valid ambiguous states. The balanced performance at $\delta = 0.1$ mirrors the theoretical "bathtub" stability, confirming that the representation space is indeed structured by a continuous potential subject to stochastic boundary noise. Full tolerance ablations and additional implementation details are deferred to Appendix J.

We further conduct an ablation study to decouple the geometric components used by the dual-stream protocol. As shown in Table 3, *R+TC* reaches 96.0% token accuracy, indicating that the model's latent local computation remains highly robust even when the final output token is wrong. In contrast, *TR+C* only reaches 90.5% token accuracy, mechanically isolating the noisy carry potential as the main bottleneck behind arithmetic failures.

## 9. Related Works

### 9.1. LLM Arithmetic and Probing

While LLMs demonstrate impressive arithmetic capabilities (Cobbe et al., 2021; Hendrycks et al., 2021; Yuan et al., 2023; Zhang et al., 2024), the underlying mechanisms remain debated. One paradigm views arithmetic as discrete symbolic manipulation, where inputs are mapped to categorical sum types (Quirke & Barez, 2024) or rely on optimized tokenization schemes (Singh & Strouse, 2024) to execute algorithmic logic. Conversely, heuristic analyses identify specific bottlenecks in this process, such as the narrow "lookahead" window for carry propagation (Baeumel et al., 2025a) and the fragility of multi-digit attention (Qiu et al., 2024). To bridge these views, recent probing studies have successfully mapped the layer-wise evolution of arithmetic signals (Yan, 2025; Zhu et al., 2025) and utilized lightweight classifiers to detect error states or decode ground truth from residual streams (Sun et al., 2025; Hedström et al., 2025). However, these approaches predominantly treat internal states as discrete variables. Our work challenges this assumption by reinterpreting arithmetic generation as the noisy quantization of a *Continuous Carry Potential*, providing a mechanistic explanation for why linear probes can

successfully disentangle these signals.

### 9.2. Geometry of Reasoning

Broader studies on Riemannian geometry and topological persistence offer a foundational backdrop (Shao et al., 2018; Naitzat et al., 2020; Fitz et al., 2024; Azizian et al., 2025; Tiblias et al., 2026), yet deciphering arithmetic requires analyzing specific algebraic structures. Nanda et al. (2023) hypothesized that models perform modular arithmetic via rotational dynamics on circular manifolds. Recent refinements suggest LLMs utilize high-dimensional spirals or trigonometric encodings (Kantamneni & Tegmark, 2025), often composing digit-wise circular representations with linear magnitude components (Levy & Geva, 2025; Engels et al., 2026). However, abstract topologies rarely map directly to concrete failure modes. Our framework refines these helical hypotheses by revealing a locally stratified manifold where the proximity of carry fibers determines error probabilities via mechanistic *geometric slippages*.

## 10. Conclusion and Discussion

This work identifies the **IRST** as the geometric backbone of arithmetic, where errors manifest as mechanistic *geometric slippages*. Our **Noisy Quantization Model** formalizes this process, showing that models often maintain a correct continuous *Carry Potential* internally but fail during discrete token selection due to neural noise near quantization boundaries. Empirically, we observe that error distributions are predominantly off-by-one ($\pm 1$ shifts). Our geometric framework provides a structural explanation: unlike a circular clock-face representation, Anchors 0 and 9 are geometrically distant along the backbone. This helical geometry imposes a high energetic barrier against wrap-around and non-unit errors, effectively restricting failure modes to local shifts along a continuous IRST fiber.

Our methodology leverages the single-digit tokenization of models (e.g., Qwen3 (Yang et al., 2025), Gemma3 (Team et al., 2025)), which allows for a direct alignment between visual digits and internal arithmetic states without the interference of multi-digit grouping. While we hypothesize that BPE-based models rely on similar latent geometries, their carry signals are likely entangled within dense multi-digit embeddings, presenting a frontier for future research. At the same time, our current evidence still combines UMAP visualization and probe-based decoding rather than a full circuit-level localization of the arithmetic computation, and our empirical focus remains mainly on addition. Even so, the success of our dual-stream consistency check confirms that hallucinations are often quantization failures of a correct continuous potential, suggesting that improving LLM reliability depends on stabilizing the mapping from these continuous neural thoughts to discrete symbolic outputs.

## Acknowledgement

This work is supported in part by the National Natural Science Foundation of China (62576160), Fundamental Research Funds for the Central Universities (KG202508), 111 Center (B26023), and Fundamental and Interdisciplinary Disciplines Breakthrough Plan of the Ministry of Education of China (JYB2025XDXM118).

## Impact Statement

This paper presents work whose goal is to advance the field of Machine Learning. There are many potential societal consequences of our work, none which we feel must be specifically highlighted here.

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

# A. Experimental Details for Section 3

## A.1. Model Configuration and Inference

We perform all experiments in the main text using the **Qwen3-4B** model (Yang et al., 2025). All generations are performed using greedy decoding (temperature $T = 0$) to eliminate stochasticity and ensure that the internal representations reflect the model's deterministic arithmetic reasoning path.

## A.2. Dataset Construction

Our dataset $\mathcal{D}$ consists of $N = 10,000$ distinct samples of 3-term addition problems ($A_0 + A_1 + A_2 = S$), where each addend is a 10-digit integer.

**Prompt Template.** To enforce deterministic arithmetic output and suppress conversational filler, we employ a *forced-prefix* strategy. We apply the model's standard chat template to the instruction, and then explicitly append the arithmetic expression and an equals sign as the start of the assistant's response. An example from our dataset is shown below:

```
Prompt and Generation Example (Real Sample)

[User Control Tokens]
Calculate 651507672825 + 369229089834 + 294552136401.  Only output a number.  Don't
output commas.

[Assistant Control Tokens]
651507672825 + 369229089834 + 294552136401 = 1315289000060
```

In this example, the ground truth sum is `1315288899060`.

**Truncated Generation Protocol.** To isolate the precise origin of arithmetic errors, we employ an *early stopping* strategy. As shown in the example above, the model correctly generates the prefix `131528` but deviates at the next position (generating `9` instead of `8`). We terminate generation immediately at this first error. This ensures that every analyzed activation $\boldsymbol{h}_p^{(L)}$ is conditioned on a strictly correct history, confirming that the observed geometric slippage is a spontaneous divergence from the correct manifold rather than a downstream effect of error propagation.

**Data Balancing.** Since random 10-digit addition yields imbalanced carry distributions, we employ stratified sampling to ensure the manifold at $p = 4$ is fully populated. We enforce a uniform distribution of $c_p \in \{0, 1, 2\}$ at this target position to prevent visualization bias. Crucially, this intervention is strictly local; the natural carry distribution is preserved across all other positions, ensuring the dataset as a whole retains its global statistical representativeness.

## A.3. Activation Extraction

**Tokenization Alignment.** We verify that the Qwen3 tokenizer treats each decimal digit as a single token in our prompt format. This one-to-one mapping between mathematical digits and semantic tokens is crucial for position-wise analysis.

**Position Indexing.** Consistent with our mathematical framework, the generation index $p \in \{0, \ldots, P - 1\}$ proceeds from the most significant digit ($p = 0$) to the least. Consequently, our primary analysis target $p = 4$ corresponds to the 5th generated token (mathematical significance $k = P - 5$). We extract the residual stream activation $\boldsymbol{h}_p^{(L)}$ from the final layer $L = 36$, taken after the final normalization layer and immediately prior to the unembedding projection.

## A.4. Dimensionality Reduction Settings

We employ UMAP (McInnes et al., 2018) to project the 2560-dimensional activation vectors into 2D. We utilize the *cosine distance* metric to capture angular semantic relationships. To preserve the global topological structure while maintaining local separation, we set the hyperparameters to `n_neighbors=300` and `min_dist=0.3`. For visual clarity in the presented figures, we apply random downsampling to the projection to mitigate overplotting in high-density regions.

# B. Comparison of Different Dimensionality Reductions

We performed a robustness check using Principal Component Analysis (PCA) (Abdi & Williams, 2010) and t-Distributed Stochastic Neighbor Embedding (t-SNE) (Maaten & Hinton, 2008) as comparisons. We used the identical dataset of final-layer hidden states ($\boldsymbol{h}_p^{(L)}$ at position $p = 4$) as presented in Section 3.

## B.1. Quantitative Evaluation

We quantitatively assess the quality of the low-dimensional embeddings using the `trustworthiness` metric (Pedregosa et al., 2011) with `n_neighbors=50`. This metric evaluates how well the local neighborhood structure of the high-dimensional space is preserved in the 2D projection. As shown in Table 4, both non-linear methods (UMAP and t-SNE) achieve near-perfect scores, significantly outperforming the linear PCA projection.

*Table 4.* Trustworthiness scores for different dimensionality reduction methods on $\boldsymbol{h}_p^{(L)}$ at $p = 4$.

| Method | Trustworthiness ($\uparrow$) |
|---|---|
| PCA (Linear) | 0.8633 |
| t-SNE (Non-linear) | 0.9907 |
| **UMAP (Non-linear, Main Text)** | **0.9953** |

## B.2. Analysis of Visualizations

Figure 7 presents the visualizations for PCA and t-SNE.

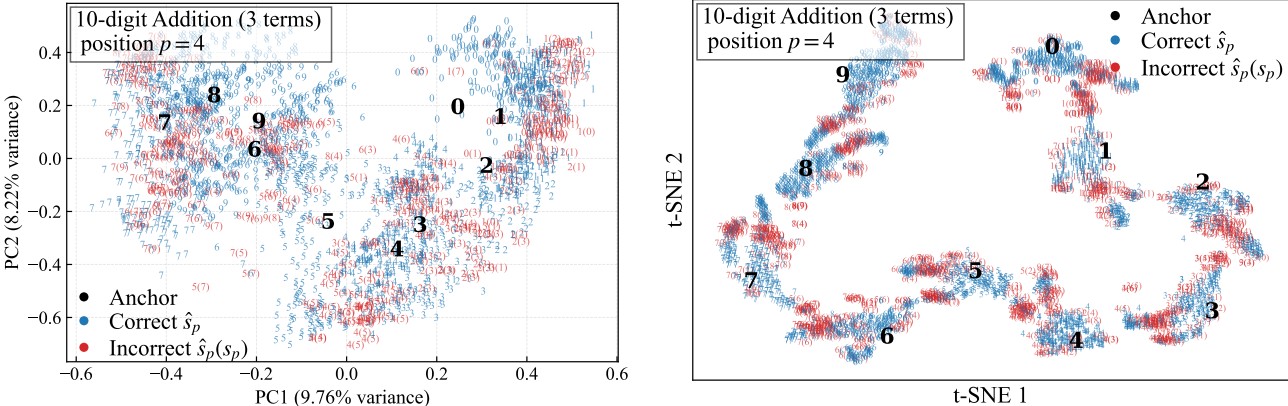

*Figure 7.* **Alternative Dimensionality Reduction Visualizations.. (Left) PCA Projection:** Only macroscopic digit clustering is visible; the fine-grained IRST fibers are collapsed due to the low variance captured by the first two components (9.76% and 8.22%). **(Right) t-SNE Projection:** The visualization clearly recovers the same structural hierarchy (Macroscopic Backbone and Microscopic Texture) as UMAP, confirming that the IRST is not an artifact of a specific algorithm.

**Linear vs. Non-linear:** The failure of PCA to reveal the backbone and texture structure is indicative of the non-linearity of the arithmetic manifold. The first two principal components explain only approximately 17.98% of the total variance, suggesting that the relevant arithmetic information is distributed across a much higher-dimensional subspace. The significantly higher trustworthiness of non-linear methods confirms that local neighborhood preservation is essential for interpreting the model's internal states.

**t-SNE vs. UMAP:** The t-SNE projection is qualitatively consistent with our UMAP results, exhibiting clear digit basins and stratified carry fibers. This cross-algorithm consistency provides strong evidence that the IRST is an intrinsic geometric feature of the residual stream. We chose UMAP for the main text because it marginally outperforms t-SNE in trustworthiness (0.9953 vs 0.9907) and better preserves global continuity, which is critical for representing trajectories that span multiple digit basins.

## C. Geometric Validation and Causal Steering

The main text already presents a representative trajectory-level case study and causal steering result for $\mathcal{T}_3$ (Figure 6). In this appendix, we restore the full trajectory-level interpretation of that figure before turning to the broader native-space ablations and intrinsic-dimensionality measurements.

### C.1. Detailed Case Study on $\mathcal{T}_3$

We first visualize the latent geometry of this trajectory. We calculate the centroids of the stable clusters for $(3, 3, 0)$, $(4, 4, 1)$, and $(5, 5, 2)$ in the last-layer ($p = 4$) representation space.

Figure 6 (Left) presents the projection of all samples belonging to $\mathcal{T}_3$. The x-axis represents the cosine distance relative to the central node centroid $(4, 4, 1)$, while the y-axis represents the analytical Carry Potential $\Phi$. The empirical distribution mirrors the theoretical schematic in the main text (Figure 3): the low-potential cluster $(3, 3, 0)$, the central cluster $(4, 4, 1)$, and the high-potential cluster $(5, 5, 2)$ are connected by a clear V-shaped progression. Crucially, leakage errors like $(5, 4, 2)$ and hallucination errors like $(4, 5, 1)$ reside precisely in the transition zones between the stable basins. This confirms that these errors are geometric intermediate states where the representation has drifted away from the cluster center but has not yet crossed the decision boundary of the next carry fiber.

### C.2. Detailed Causal Steering Dynamics

Using the same steering intervention defined in Section 6, we can interpret Figure 6 (Right) at the level of individual thresholds. The four representative samples are ordered exactly as their geometric positions would suggest:

- **Stable State** $(5, 5, 2)$ ($\Phi = 2.36$): This sample is deeply embedded in the "Carry 2" basin. It requires a significant negative steering force ($\alpha \approx -0.5$) to suppress the carry signal and flip the output back to 4.

- **Hallucination Error** $(4, 5, 1)$ ($\Phi = 1.96$): This sample represents a "barely" wrong 5. It lies extremely close to the decision boundary and requires only a minimal push ($\alpha \approx -0.1$) to correct it back to the ground truth 4.

- **Leakage Error** $(5, 4, 2)$ ($\Phi = 2.00$): This sample represents a "barely" wrong 4. It resides just below the threshold and is corrected to 5 with a positive push of $\alpha \approx 0.3$.

- **Stable State** $(4, 4, 1)$ ($\Phi = 1.62$): This sample is stable in the "Carry 1" basin. It requires a strong positive force ($\alpha \approx 0.5$) to induce a transition to 5.

The precise ordering of these transition thresholds quantitatively confirms that the model's arithmetic decision is strictly determined by the projection of the representation onto the Carry Potential direction.

### C.3. Native-Space Validation Across Trajectories

As an ablation on the projection choice, we also measure the trajectory geometry directly in the original residual space. Figure 8 shows that the native-space cosine-distance profiles preserve the same V-shaped correlation with $\Phi$ across $\mathcal{T}_0 \ldots \mathcal{T}_9$. In other words, the core ordering is already present in $\mathbb{R}^{2560}$ and does not depend on a particular 2D UMAP layout.

### C.4. Intrinsic-Dimensionality Analysis

Figure 9 summarizes intrinsic-dimensionality estimates computed directly in native space. Across $\mathcal{T}_0 \ldots \mathcal{T}_9$, the trajectory-conditioned subsets remain consistently low-dimensional, while the pooled set exhibits a noticeably larger linear effective dimension. This pattern supports the picture that individual IRSTs occupy relatively compact geometric subsets embedded within a broader shared arithmetic scaffold.

As further illustrated in Figure 10, we additionally examined layer-wise intrinsic-dimension estimates for $\mathcal{T}_5$ and the pooled set (All). The two are broadly similar before layer 23. Between layers 23–31, $\mathcal{T}_5$ has slightly lower nonlinear ID (MLE) than All, while after layer 31 this trend reverses. One possible reading is that a relatively stable global scaffold forms first, followed by later refinement of trajectory-specific local structure, broadly consistent with Appendix K.

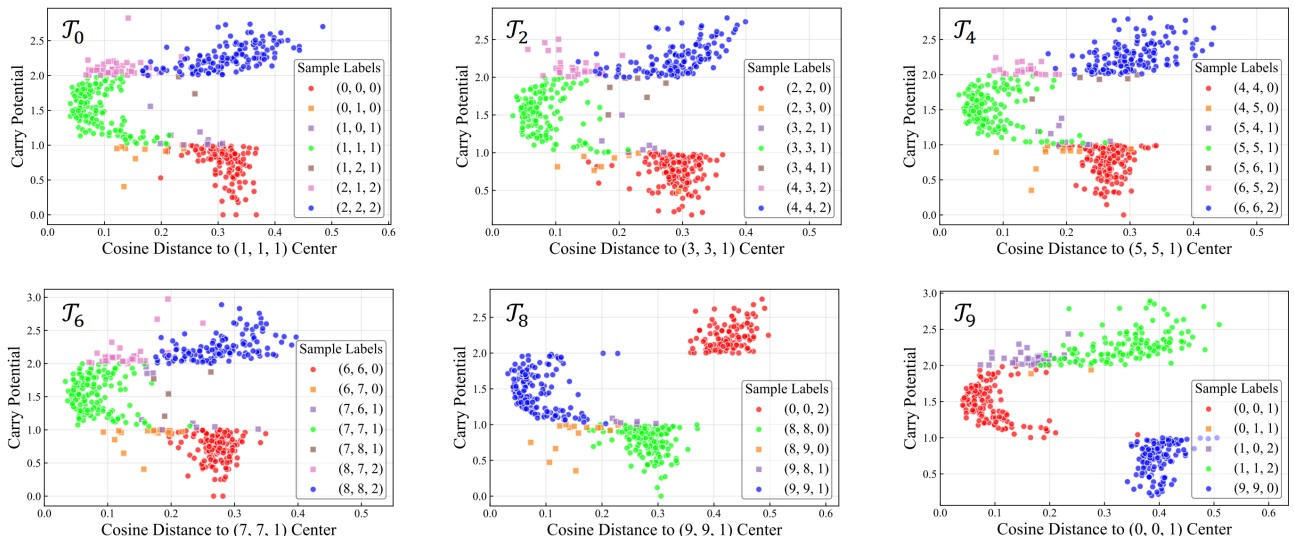

*Figure 8.* **Projection-independent validation of the IRSTs in the native representation space.** Cosine distance to anchor states is evaluated directly in $\mathbb{R}^{2560}$ across multiple trajectories. The characteristic V-shaped correlation between native-space distance and the continuous carry potential $\Phi$ generalizes across $\mathcal{T}_0 \ldots \mathcal{T}_9$, supporting the claim that the IRST organization is not solely a 2D projection artifact. The exceptional behavior near $\mathcal{T}_8$ and $\mathcal{T}_9$ is consistent with the absence of a continuous adjacent-digit bridge across the $0 \leftrightarrow 9$ boundary.

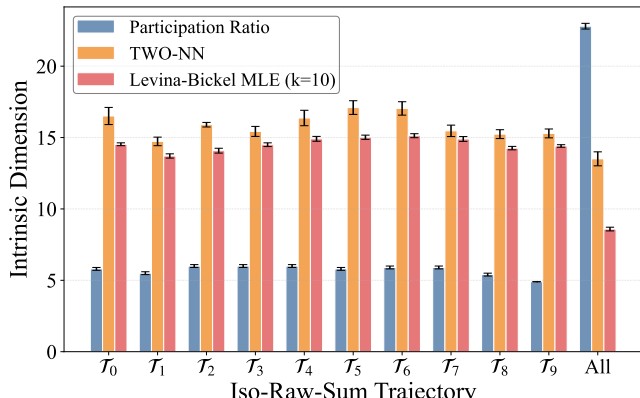

*Figure 9.* **Intrinsic dimensionality across IRSTs.** We report the participation ratio, TWO-NN, and Levina–Bickel MLE estimates for $\mathcal{T}_0 \ldots \mathcal{T}_9$ and their pooled union. The trajectory-conditioned subsets remain stably low-dimensional in native space, while the pooled set exhibits a larger linear effective dimension.

## D. Validation of the Raw Sum Assumption

In the main context, we posit that the model's arithmetic errors are primarily driven by latent carry deviations rather than failures in local column-wise summation. This assumption, mathematically expressed as $\hat{r}_p \approx r_p$, is foundational to both our theoretical derivation of latent carry states and the design of the inference-time correction mechanism.

To validate this, we trained an 3-layer MLP classification probe to predict the absolute raw sum $r_p = \sum_i a_{i,p}$ from the final layer activations $\boldsymbol{h}_p^{(L)}$ at $p = 4$. For our 3-term addition task, $r_p$ ranges from 0 to 27, constituting a 28-class classification problem. Crucially, we evaluated this probe on a balanced dataset containing equal numbers of correct and incorrect model generations.

**Results.** The probe achieved an Accuracy of **89.35%** and an AUC of **99.59%** on the validation set, far exceeding the random baseline (Accuracy $\sim 3.6\%$). While this performance is not perfect—implying that a minority of errors may indeed stem from genuine local calculation failures—the high fidelity supports two critical conclusions.

On the theoretical front, it indicates that the model predominantly encodes the correct local sum even when it generates

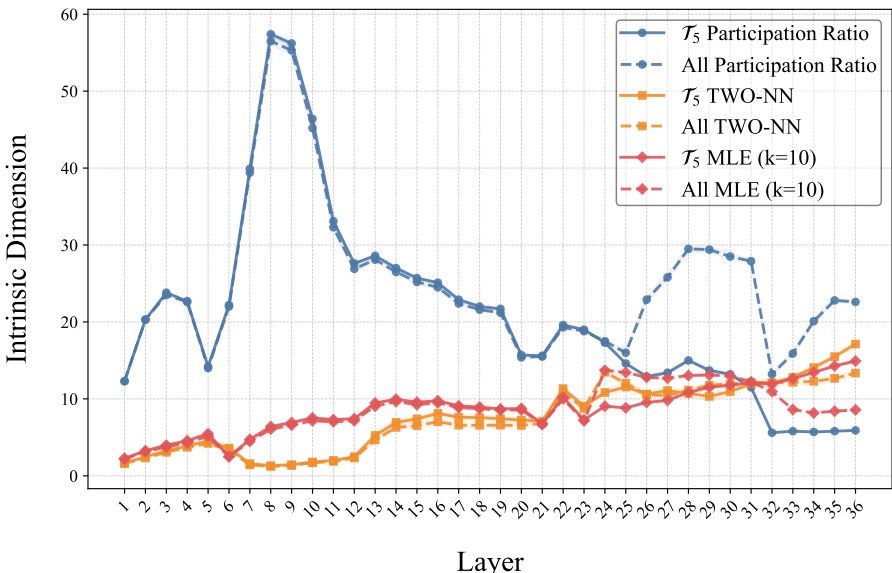

*Figure 10.* **Layer-wise evolution of intrinsic dimensionality.** We compare the nonlinear and linear intrinsic-dimension estimates for $\mathcal{T}_5$ and the pooled set of all trajectories across layers. The two remain broadly similar in early layers, while later layers develop more trajectory-specific local structure.

an incorrect output digit $\hat{s}_p$. This suggests that arithmetic errors are largely geometrically constrained: the representation typically remains anchored to the correct **IRST** (preserving $r_p$) but slides along it into an incorrect carry fiber. Consequently, mapping output errors to carry deviations ($\hat{c}_p = \hat{s}_p - r_p$) is physically justified as the dominant error mechanism, though not an exclusive one. On the methodological side, the robustness of the internal raw sum signal validates the *Local Calculation Stream* in our dual-stream intervention method (Section 8), ensuring that extracting $\hat{r}_p$ serves as a dependable, albeit probabilistic, anchor for reconstruction in the vast majority of instances.

This distinction is visualized directly in Figure 11. The carry-based subset follows the same boundary-concentrated pattern emphasized in the main text, whereas the raw-sum-failure subset is geometrically far less structured.

## E. Derivation of Latent Carry States

In this section, we provide the detailed derivation for inferring the latent predicted input carry $\hat{c}_p$ from the observed model output $\hat{s}_p$. This inference is grounded in the modular arithmetic properties of the addition task.

**Problem Setup.** Recall the modular relationship for the ground truth digit $s_p$, derived from the raw sum $r_p$ and the ground truth input carry $c_p$:

$$s_p \equiv (r_p + c_p) \pmod{10}. \tag{10}$$

We posit that the model generates its predicted digit $\hat{s}_p$ using the same logic but based on its internal states. Assuming the local summation is robust (i.e., the perceived raw sum matches the true raw sum $\hat{r}_p = r_p$), any deviation in the output must stem from a latent predicted carry $\hat{c}_p$:

$$\hat{s}_p \equiv (r_p + \hat{c}_p) \pmod{10}. \tag{11}$$

We analyze the carry implications for the two primary error modes: Underestimation and Overestimation.

**1. Underestimation (Leakage).** This error mode occurs when the model outputs a digit that is exactly 1 less than the ground truth (modulo 10). Mathematically, this is expressed as:

$$\hat{s}_p \equiv (s_p - 1) \pmod{10}. \tag{12}$$

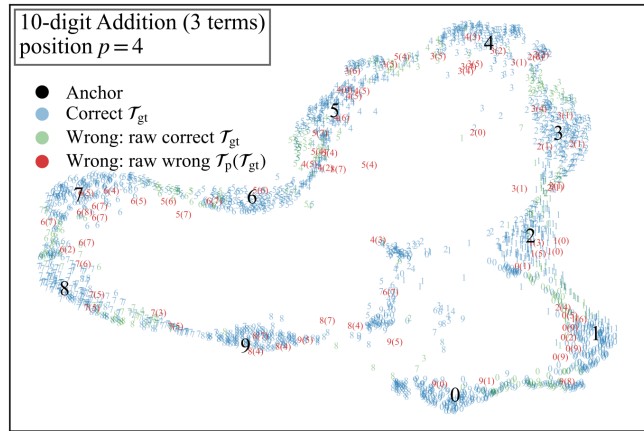

*Figure 11.* **Geometric signatures of carry-based and non-carry errors.** Error modes are decoupled using the raw-sum probe. Carry-based errors, where the local raw sum is still recovered correctly, remain concentrated near adjacent decision boundaries. In contrast, non-carry errors scatter across more distant regions and do not follow a single continuous trajectory, indicating a distinct failure mode beyond the dominant carry-deviation mechanism. In the plotted annotations, $\mathcal{T}$ denotes the inferred trajectory identity, $gt$ denotes the ground-truth digit, and $p$ denotes the analyzed generation position (fixed here at $p = 4$).

Substituting the definitions from Equation (10) and Equation (11) into this relation:

$$(r_p + \hat{c}_p) \equiv (r_p + c_p - 1) \pmod{10}$$
$$\hat{c}_p \equiv (c_p - 1) \pmod{10}. \tag{13}$$

Since carry values in standard addition are small integers (typically $c_p \in \{0, 1, 2\}$ for 3 terms addition), this equivalence implies $\hat{c}_p = c_p - 1$. Thus, an output underestimation directly maps to a *Leakage* of the internal carry.

**2. Overestimation (Hallucination).** This error mode occurs when the model outputs a digit that is exactly 1 greater than the ground truth (modulo 10):

$$\hat{s}_p \equiv (s_p + 1) \pmod{10}. \tag{14}$$

Substituting the definitions yields:

$$(r_p + \hat{c}_p) \equiv (r_p + c_p + 1) \pmod{10}$$
$$\hat{c}_p \equiv (c_p + 1) \pmod{10}. \tag{15}$$

This implies $\hat{c}_p = c_p + 1$. Thus, an output overestimation maps to a *Hallucination* of an additional carry unit.

## F. Derivation of the Noisy Quantization Error Rate

We derive the error probability for the **Noisy Quantization Model**. Let $\Phi$ be the ground truth potential and $\hat{\Phi} = \Phi + \epsilon$ be the noisy estimate with $\epsilon \sim \mathcal{N}(0, \sigma^2)$. The discrete carry is $\hat{c} = \lfloor \hat{\Phi} \rfloor$. We define the fractional part as $\delta(\Phi) = \Phi - \lfloor \Phi \rfloor$. The standard Q-function is defined as:

$$Q(z) = \frac{1}{\sqrt{2\pi}} \int_z^\infty e^{-t^2/2} \, dt. \tag{16}$$

**1. Underestimation (Leakage).** Leakage occurs when the noise pushes the potential below the integer floor, i.e., $\Phi + \epsilon < \lfloor \Phi \rfloor$, which simplifies to $\epsilon < -\delta(\Phi)$. Utilizing the symmetry of the Gaussian distribution ($P(Z < -z) = Q(z)$), the probability is:

$$P_{\text{leak}}(\Phi) = P\left(\frac{\epsilon}{\sigma} < -\frac{\delta(\Phi)}{\sigma}\right) = Q\left(\frac{\delta(\Phi)}{\sigma}\right). \tag{17}$$

**2. Overestimation (Hallucination).** Hallucination occurs when the noise pushes the potential above the next integer ceiling, i.e., $\Phi + \epsilon \geq \lfloor \Phi \rfloor + 1$, which implies $\epsilon \geq 1 - \delta(\Phi)$. The probability is directly given by the Q-function:

$$P_{\text{halluc}}(\Phi) = P\left(\frac{\epsilon}{\sigma} \geq \frac{1 - \delta(\Phi)}{\sigma}\right) = Q\left(\frac{1 - \delta(\Phi)}{\sigma}\right). \tag{18}$$

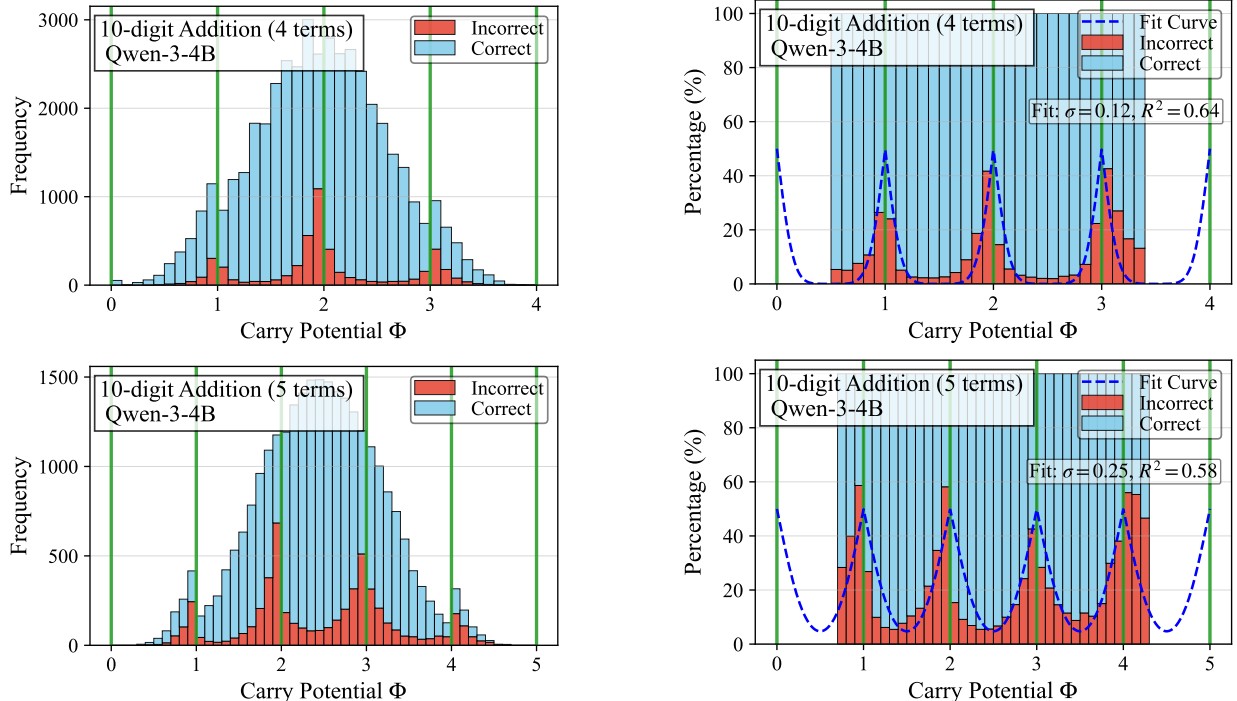

*Figure 12.* **Scaling of cognitive noise with task complexity.** We extend the analysis of Figure 5 to 4-term ($n = 4$) and 5-term ($n = 5$) addition. The periodic bathtub structure persists across settings, validating the universality of the quantization mechanism. However, the fitted noise level $\sigma$ increases significantly from 0.05 ($n = 3$) to 0.12 ($n = 4$) and 0.25 ($n = 5$), indicating a sharp degradation in signal fidelity as the operand count grows.

**Total Error Probability.** Assuming negligible probabilities for non-unit errors (valid for small $\sigma$), the total error rate is the sum of the leakage and hallucination risks:

$$P(\text{Error} \mid \Phi) = Q\left(\frac{\delta(\Phi)}{\sigma}\right) + Q\left(\frac{1 - \delta(\Phi)}{\sigma}\right). \tag{19}$$

## G. Scaling Dynamics of Cognitive Noise

We extend our analysis to higher-complexity tasks to investigate how the cognitive noise $\sigma$ scales. As shown in Figure 12, the periodic *bathtub* error distribution is preserved in both 4-term and 5-term addition, confirming that the quantization of a continuous carry potential is a universal mechanistic primitive in LLM arithmetic. Notably, we observe a sharp non-linear growth in $\sigma$ as the number of addends ($n$) increases: from $\sigma \approx 0.05$ at $n = 3$ to $\sigma \approx 0.25$ at $n = 5$. This increase in noise causes the overlap of Gaussian tails near the quantization thresholds, effectively eliminating the *robust plateaus* where error rates were previously near zero.

Our analysis identifies three primary drivers of this noise. First, the *operand count (n)* increases the informational load on the attention mechanism, diluting the signal-to-noise ratio during the aggregation of carry potential. Second, longer *digit lengths (m)* incur a cumulative cognitive load, leading to higher baseline noise. Finally, while the results in the main text aggregate all positions to demonstrate the general mechanism, a boundary-focused analysis reveals that the edge positions ($p = 0$ and $p = 9$) deviate from the interior periodic regime. As shown in Figure 13, these boundary columns exhibit distinct frequency and error profiles, consistent with reduced effective carry ambiguity at the most significant digit and stricter discrete constraints at the least significant digit. These findings suggest that the arithmetic capacity of Transformers is fundamentally bounded by the accumulation of this analog neural noise, a scaling property we leave for future systematic formulation.

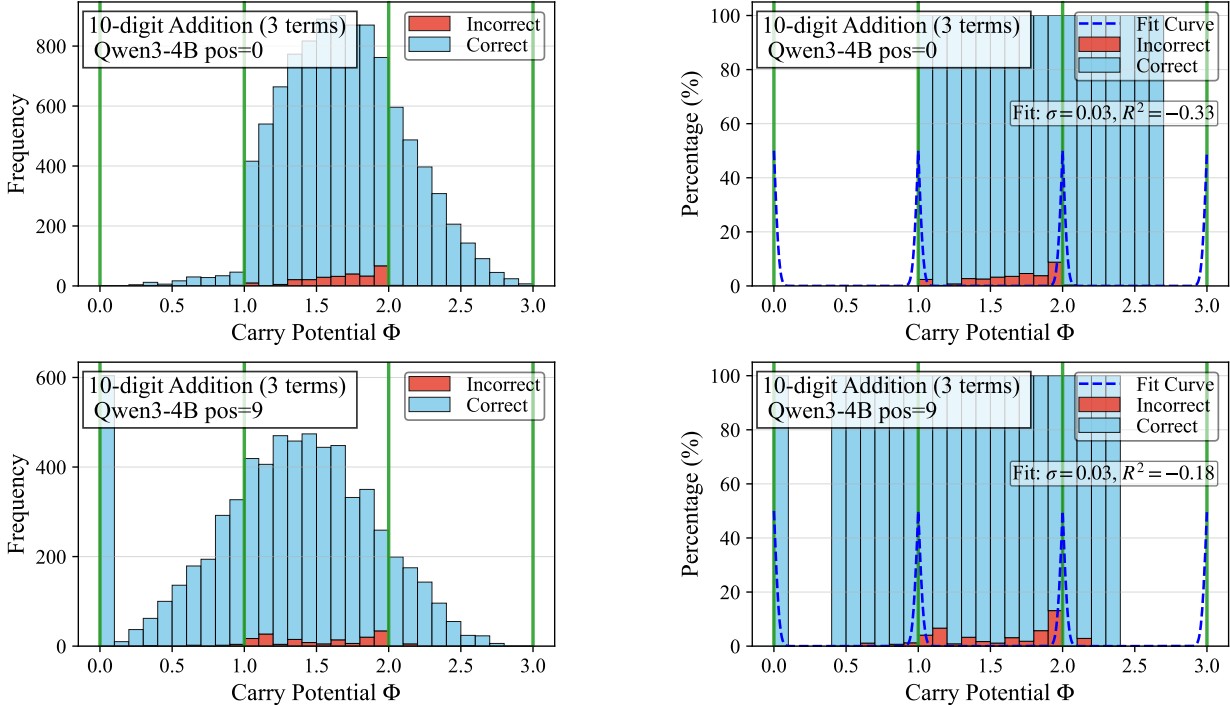

*Figure 13.* **Boundary-position bathtub profiles.** Frequency and conditional-error distributions at the most significant digit ($p = 0$, top row) and the least significant digit ($p = 9$, bottom row) for 3-term 10-digit addition. Unlike the interior columns summarized in the main text, these boundary positions show structurally different profiles, highlighting that the steady-state bathtub regime is primarily an interior-position phenomenon.

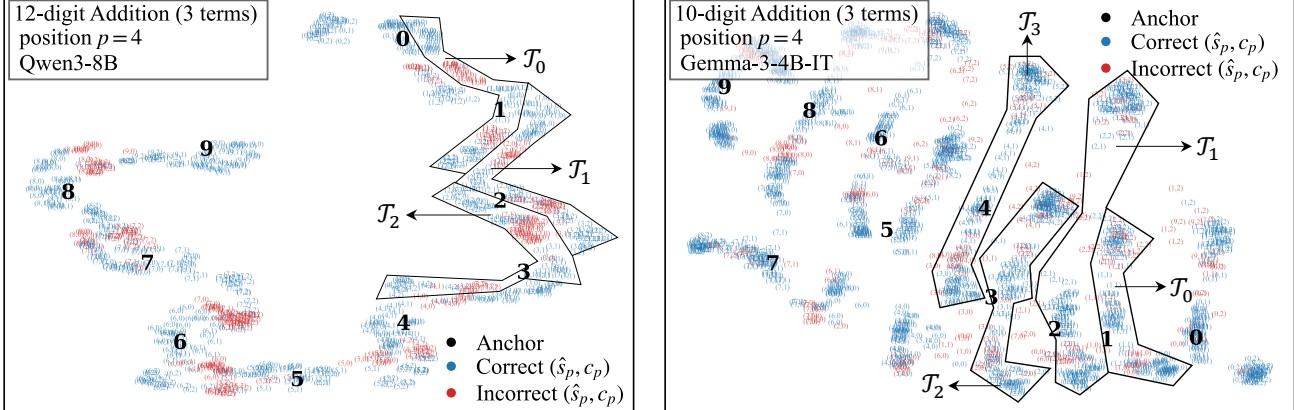

*Figure 14.* **Generalization of the IRST geometry across different models. (Left)** UMAP visualization for **Qwen3-8B** on a 12-digit addition task. The manifold structure is highly consistent with the 4B model, featuring a sequential arrangement of digit basins (0–9) connected by clear trajectories (e.g., $\mathcal{T}_0, \mathcal{T}_1, \mathcal{T}_2$). **(Right)** UMAP visualization for **Gemma-3-4B-IT** on a 10-digit addition task. Despite architectural differences, the fundamental geometry of IRSTs remains (e.g., $\mathcal{T}_0 \sim \mathcal{T}_3$).

# H. Generalize to Other LLMs

To validate the universality of our geometric framework, we extend our analysis beyond the Qwen3-4B model used in the main text. We visualize the last-layer representations of two additional models with different scales and architectures: **Qwen3-8B** and **Gemma-3-4B-IT**.

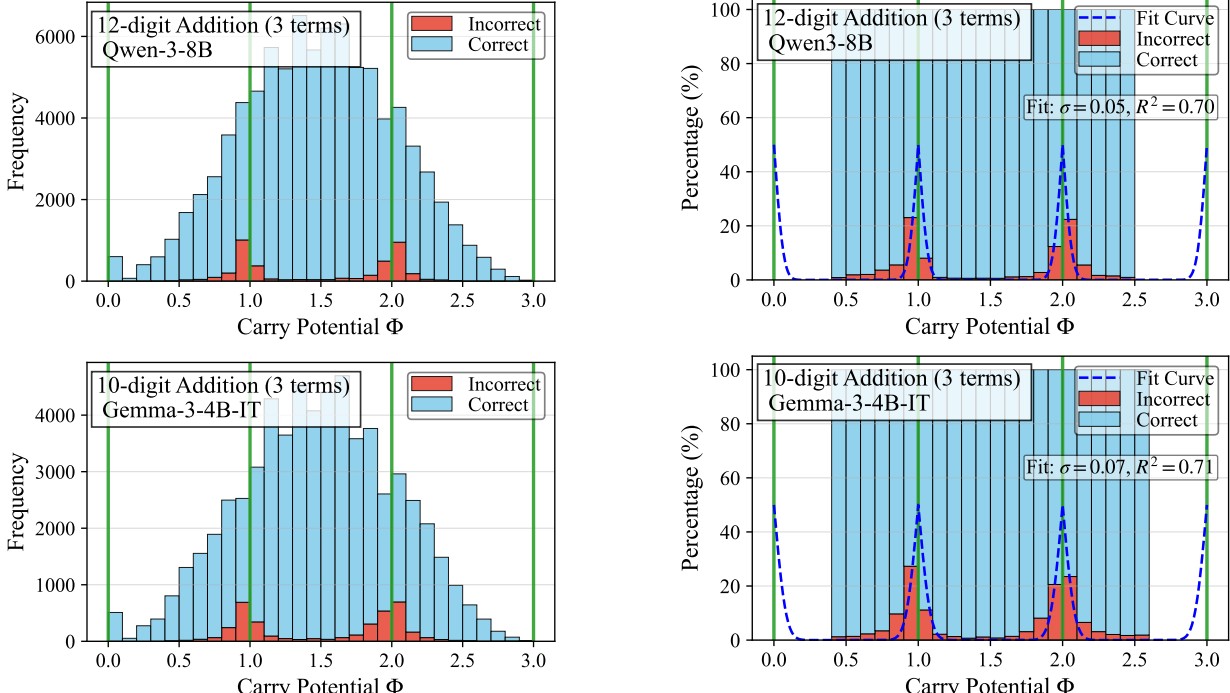

*Figure 15.* **Generalization of the Noisy Quantization hypothesis.** Validation on Qwen3-8B (12-digit addition, **top**) and Gemma-3-4B-IT (10-digit addition, **bottom**). The **Left** panels show the sample frequency distribution relative to Carry Potential $\Phi$, indicating that the dataset covers the entire potential space. The **Right** panels display the conditional error rates (red bars) overlaid with our theoretical fit (dashed blue line). The recurrence of the periodic error pattern near integer boundaries (green lines) and the high fitting scores ($R^2 \approx 0.7$) confirm that the geometric quantization mechanism is robust across different model sizes and families.

### H.1. IRSTs

As shown in Figure 14 (Left), the Qwen3-8B model exhibits a manifold geometry strikingly similar to its smaller counterpart Qwen3-4B in Figure 2. The digit anchors are arranged in a distinct, quasi-linear sequence from 0 to 9. The IRSTs form continuous paths that seamlessly bridge adjacent digit basins. This indicates that the geometric encoding strategy is stable across model scales within the same architectural lineage.

The Gemma-3-4B-IT model (Figure 14, Right), representing a different model family, presents a slightly different global curvature. However, the core geometric properties predicted by our framework remain intact. We can clearly identify the basin structure for digits 0–9 and trace specific IRSTs.

This picture is further supported by the **top** panel of Figure 16, which visualizes the single-task arithmetic transformers of Quirke et al. (2025). The under-converged checkpoint still exhibits continuous inter-basin bridges similar to the IRST geometry in our main setting, whereas the fully converged checkpoint resolves into more isolated basins. This suggests that the same continuous organization can arise as an intermediate representational regime even in specialized arithmetic models.

These observations suggest that our IRST is not an artifact of a specific model checkpoint but likely a convergent mechanism for arithmetic representation in state-of-the-art LLMs. Regardless of the specific architecture, models appear to learn to organize arithmetic states by "threading" raw-sum manifolds through semantic digit anchors.

### H.2. Noisy Quantization Dynamics

We further validate the dynamic mechanism of error generation—the Noisy Quantization Model—across different model families. Figure 15 visualizes the relationship between the continuous Carry Potential $\Phi$ and the generation accuracy for Qwen3-8B and Gemma-3-4B-IT, while Figure 16 extends the comparison to specialized arithmetic transformers trained solely on addition. Together, these results show that the same quantization-driven error landscape appears both in

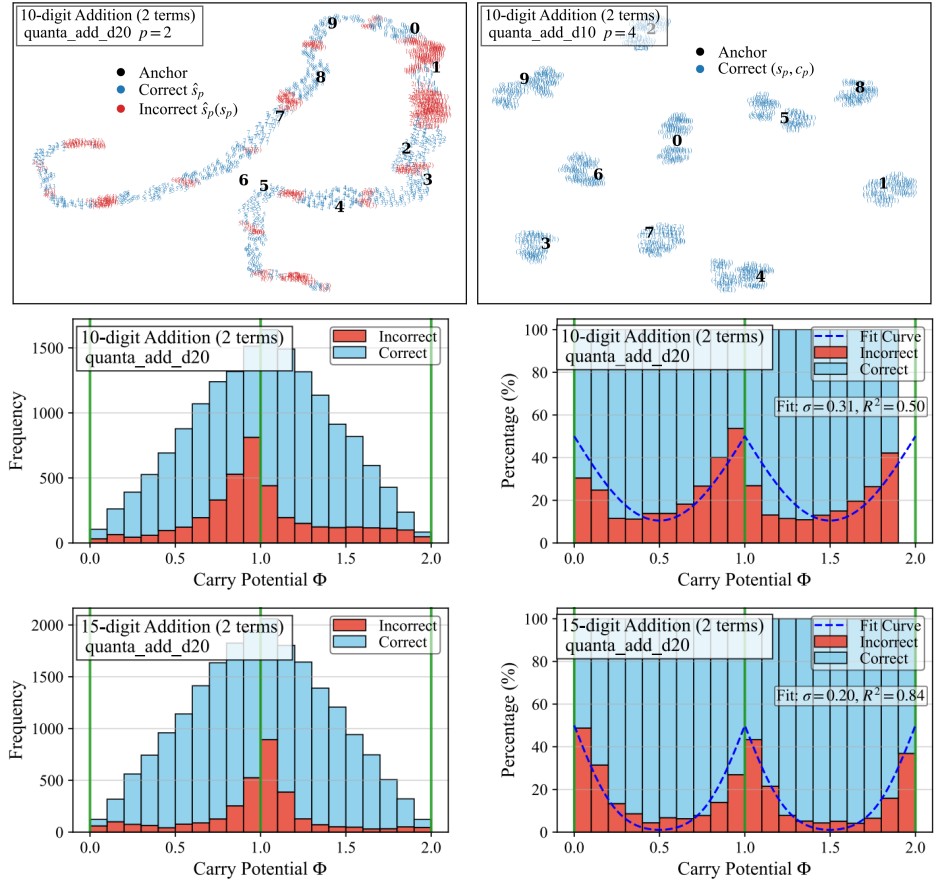

*Figure 16.* **Additional evidence from specialized arithmetic transformers. (Top)** UMAP projections from the under-converged and fully converged single-task addition models reported by Quirke et al. (2025). The under-converged model exhibits continuous inter-basin geometry similar to our main setting, while the fully converged model forms more isolated basins. **(Bottom)** Conditional error-rate curves for the under-converged model still follow the periodic bathtub pattern predicted by the Noisy Quantization Model.

general-purpose LLMs and in task-specific arithmetic models before full convergence.

**Recurrence of the Bathtub Curve.** As shown in the right panels of Figure 15, both Qwen-3-8B and Gemma-3-4B-IT exhibit the characteristic periodic error distribution observed in the main text. Error rates remain negligible near stable midpoints (e.g., $0.5$, $1.5$) but spike sharply as $\Phi$ approaches integer boundaries ($1.0$, $2.0$). This confirms that arithmetic failures are structurally induced by signal ambiguity near quantization thresholds rather than random stochasticity. The **bottom** panel of Figure 16 shows the same bathtub-shaped error profile in the under-converged specialized arithmetic transformer, indicating that geometric slippage is not unique to general-purpose LLMs.

**Theoretical Fit and Noise Levels.** Fitting our theoretical error function (Equation (6)) to the empirical data yields a robust fit ($R^2 \geq 0.70$) for both models. We estimate the cognitive noise levels to be $\sigma \approx 0.05$ for Qwen-3-8B and $\sigma \approx 0.07$ for Gemma-3-4B-IT. Notably, Qwen-3-8B is evaluated on a more challenging 12-digit addition task compared to the 10-digit task used for the 4B models. The fact that Qwen-3-8B maintains a similarly low noise level ($\sigma \approx 0.05$) despite the increased sequence length indicates superior arithmetic precision, as the smaller model would likely exhibit significantly higher noise if subjected to this harder regime. In the same spirit, the specialized arithmetic transformers in Figure 16 provide a qualitative training-time contrast: the under-converged checkpoint still exhibits the bathtub-shaped failure profile, whereas the fully converged checkpoint shows more isolated basins in representation space, suggesting that full convergence sharpens the quantization geometry and suppresses boundary ambiguity. Ultimately, the universal emergence of the *bathtub* curve suggests that the Noisy Quantization Model is a fundamental phenomenological account of arithmetic processing in LLMs.

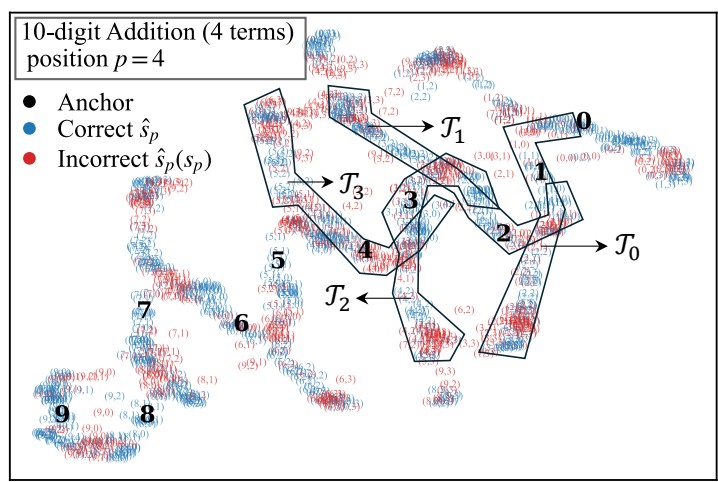 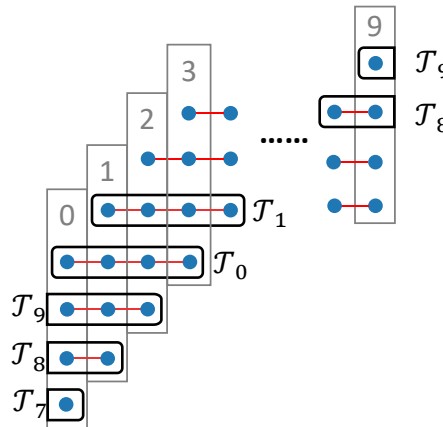

*Figure 17.* **IRST analysis for 4-term addition** ($c_p \in \{0, 1, 2, 3\}$). **(Left)** 2D UMAP projection of activations at $p = 4$. While the digit backbone (0-9) persists, the increased density of carry fibers causes visual entanglement of trajectories in 2D space. However 3D plots can resolve these overlaps. **(Right)** Schematic of the expanded geometry. Between any adjacent digit basins (e.g., 0 and 1), there are now three distinct transition paths (red lines) corresponding to different raw sums, representing the increased risk of carry confusion.

## I. Generalize to 4-Term Addition

In the main text, we focused on 3-term addition where the input carry $c_p$ is limited to $\{0, 1, 2\}$. To demonstrate the universality of the IRST framework, we extend our analysis to the addition of four 10-digit integers. In this setting, the maximum possible carry is 3, meaning the internal state must encode four distinct carry fibers $c_p \in \{0, 1, 2, 3\}$ within each digit basin.

### I.1. Visualizing Higher-Dimensional Manifolds

Figure 17 (Left) presents the UMAP projection of the final layer activations ($p = 4$) for the 4-term task. While the macroscopic backbone which is defined by the sequence of digit anchors $0 \rightarrow 9$ remains intact, the microscopic texture becomes significantly denser compared to the 3-term case.

We observe that distinct IRSTs (e.g., $\mathcal{T}_0, \mathcal{T}_1, \mathcal{T}_3$) still manifest as continuous paths connecting correct and incorrect states across digit boundaries. However, the 2D projection exhibits noticeable entanglement, where different trajectories appear to cross or overlap. We attribute this to projection artifacts. The arithmetic manifold for 4-term addition possesses a higher intrinsic dimensionality required to orthogonally separate four carry states. When compressed into 2D, these dimensions collapse, creating visual occlusions.

However, these apparent overlaps disappear in a 3D UMAP projection, confirming that the fibers remain geometrically distinct in the high-dimensional representation space.

### I.2. Geometry of Error Scenarios

The schematic in Figure 17 (Right) illustrates the logical structure of this expanded manifold. The expansion of the carry range from $\{0, 1, 2\}$ to $\{0, 1, 2, 3\}$ increases the number of bridges connecting adjacent digit basins.

For any two adjacent digit anchors $k$ and $k + 1$, there are now three distinct IRSTs that traverse the boundary, corresponding to three potential error scenarios:

1. **Low Carry Confusion** ($0 \leftrightarrow 1$): Occurs on trajectory $\mathcal{T}_k$. The model confuses state $(k, c = 0)$ with $(k + 1, c = 1)$.

2. **Mid Carry Confusion** ($1 \leftrightarrow 2$): Occurs on trajectory $\mathcal{T}_{k-1}$. The model confuses state $(k, c = 1)$ with $(k + 1, c = 2)$.

3. **High Carry Confusion** ($2 \leftrightarrow 3$): Occurs on trajectory $\mathcal{T}_{k-2}$. The model confuses state $(k, c = 2)$ with $(k + 1, c = 3)$.

This confirms that the Noisy Quantization Model scales with task complexity. As the number of operands increases, the

robust plateau between quantization thresholds shrinks (packing more fibers into the same activation norm), statistically increasing the probability of geometric slippage (see Figure 12).

---

**Algorithm 1** Inference-Time Correction via Dual-Stream Consistency

---

1: **Input:** Activation $h_p^{(L)}$, Current Prediction $\hat{s}_p$, Raw Sum Probe $f_{\theta_r}$, Potential Probe $f_{\theta_\phi}$, Tolerance $\delta$
2: **Output:** Final Token $s_{final}$
3: {*1. Dual-Stream Decoding*}
4: $\hat{r}_p \leftarrow f_{\theta_r}(h_p^{(L)})$ {Decode Local Calculation Stream}
5: $\hat{\Phi}_p \leftarrow f_{\theta_\phi}(h_p^{(L)})$ {Decode Global Context Stream}
6: {*2. Construct Plausible Carry Set*}
7: $\mathcal{K}_p(\delta) \leftarrow \{\lfloor\phi\rfloor \mid \phi \in [\hat{\Phi}_p - \delta, \hat{\Phi}_p + \delta]\}$ {$\delta$-neighborhood of potential}
8: {*3. Consistency Verification*}
9: $is\_consistent \leftarrow$ False
10: **if** $\exists c \in \mathcal{K}_p(\delta)$ s.t. $\hat{s}_p \equiv (\hat{r}_p + c) \pmod{10}$ **then**
11:     $is\_consistent \leftarrow$ True
12: **end if**
13: {*4. Intervention Decision*}
14: **if** $is\_consistent$ **then**
15:     $s_{final} \leftarrow \hat{s}_p$ {Preserve original output}
16: **else**
17:     $s_{final} \leftarrow (\hat{r}_p + \lfloor\hat{\Phi}_p\rfloor) \pmod{10}$ {Intervene}
18: **end if**
19: **return** $s_{final}$

---

## J. Details for Inference-Time Correction Method

### J.1. Experiment Settings

We utilize Qwen3-4B, Qwen3-8B and Gemma-3-4B-IT to conduct the experiments. The dataset consists of $10,000$ addition problems involving 3 numbers, where each number is a positive integer with exactly 10 digits. We split the dataset into training, validation, and test sets with a ratio of $0.8 : 0.1 : 0.1$. To quantitatively evaluate the efficacy of our method, the main text focuses on three primary metrics: **Token Accuracy (Token Acc)**; **True Positive Correction (TP)**, which measures the proportion of originally incorrect tokens successfully rectified by the method; and **False Positive Preservation (FP)**, which measures the proportion of originally correct tokens that remain unchanged (following (Sun et al., 2025)). The complete implementation details of our proposed inference-time correction protocol are summarized in Algorithm 1. Unless otherwise noted, all probes described below are trained on the hidden states extracted from the final layer combing all positions. All experiments were conducted on a single NVIDIA GeForce RTX 3090 GPU. Our experiments were implemented using PyTorch (Paszke et al., 2019) and HuggingFace Transformers (Wolf et al., 2020). All models were run with `bfloat16` precision for efficient memory usage.

**Dual-Stream Probe**    The raw sum probe $f_{\theta_r}$ is a classification MLP trained to minimize Cross-Entropy loss against the ground truth $r_p \pmod{10}$. The architecture consists of an input layer, two hidden layers of 512 and 128 dimensions respectively, and an output layer projecting to 10 classes. The carry potential probe $f_{\theta_\phi}$ is a regression MLP with a similar architecture but with a scalar output. It is trained via MSE loss to estimate the continuous potential $\Phi_p$. Both probes are optimized using the Adam optimizer with a learning rate of $10^{-3}$ and early stopping based on validation accuracy (for $f_{\theta_r}$) or validation MSE (for $f_{\theta_\phi}$).

**Replacement**    This method utilizes an MLP (same architecture as $f_{\theta_r}$) to predict the ground truth digit and directly replaces the model's generation with the MLP's prediction during inference.

**Re-prompting (Sun et al., 2025)**    This method utilizes the probe as an error detector to trigger a natural language correction mechanism. The architecture of the probe is identical to that in the Replacement method. If the model's generated token matches the probe's prediction, it is appended to {`current_output`}. If not, we append a specific correction suffix to the

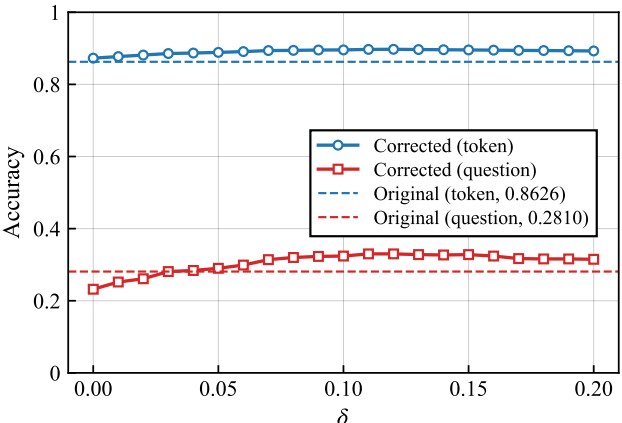

*Figure 18.* **Impact of Tolerance $\delta$.** Token accuracy (left axis, blue) and Question accuracy (right axis, orange) under different values of $\delta$. The peak at $\delta \approx 0.12$ validates the existence of a noise margin in the model's carry potential estimation.

context: *"That step looks incorrect. Let's re-do just this step: {expression} = {current_output}"*. The model is then forced to regenerate the current digit based on this augmented context. Our evaluation reveals performance degradation compared to the baseline. We attribute this to KV-cache representation locking, where erroneous vector states persist and bias attention despite lexical corrections, and to asymmetric decision boundaries, which render high-confidence errors resistant to weak intervention signals while introducing new noise during regeneration.

**Steering**  Steering guided by probe (von Rütte et al., 2024; Taufeeque et al., 2024; Cheng et al., 2024; Bhalla et al., 2024) is an emerging alternative technique. In this method, we first train a linear classification probe $P$. For a given hidden state $\boldsymbol{h}_p^{(L)}$, the probe predicts a target class $\hat{c} = \arg\max(W\boldsymbol{h}_p^{(L)})$, where $W$ is the weight matrix of $P$. The steering vector $\boldsymbol{v}$ is defined as the row in $W$ corresponding to class $\hat{c}$ (Mallen et al., 2024; Li et al., 2023). The activation is then modified as $\boldsymbol{h'}_p^{(L)} = \boldsymbol{h}_p^{(L)} + \lambda \cdot \boldsymbol{v}$. The steering strength $\lambda$ is selected via grid search over $\{0.0, 0.25, 0.5, 0.75, 1.0\}$ to maximize sample-level accuracy on the validation set.

### J.2. Further Research on Tolerance

The set of plausible carries $\mathcal{K}_p(\delta)$ effectively defines a confidence interval in the carry space:

$$\mathcal{K}_p(\delta) = \left\{ \lfloor \phi \rfloor \mid \phi \in [\hat{\Phi}_p - \delta, \hat{\Phi}_p + \delta] \right\}, \tag{20}$$

where $\delta$ is a hyperparameter representing tolerance.

To investigate the influence of $\delta$ on both token and question accuracies, we conducted inference-time correction experiments by varying $\delta$ from 0 to 0.2 in increments of 0.01. We evaluated performance using two metrics: **Token Accuracy**, which measures the proportion of individual digits correctly generated across all positions, and **Question Accuracy**, a stricter metric that counts a prediction as correct only if the entire generated number string perfectly matches the ground truth.

As illustrated in Figure 18, both token and question accuracies exhibit a non-monotonic trend, initially ascending before subsequently declining. The metrics peak simultaneously at $\delta = 0.12$, achieving a token accuracy of 0.8973 and a question accuracy of 0.3300, while the minimum performance is recorded at $\delta = 0$ (Token Acc: 0.8727, Ques Acc: 0.2320). Notably, the pronounced fluctuation observed in question accuracy aligns with the **Noisy Quantization Model**: strictly enforcing consistency ($\delta = 0$) introduces false corrections near quantization boundaries, whereas a moderate tolerance ($\delta \approx 0.1$) effectively filters out this noise while correcting genuine geometric slippages.

The set of plausible carries $\mathcal{K}_p(\delta)$ therefore functions as a narrow confidence interval in carry space, allowing the method to preserve ambiguous but still plausible internal states rather than forcing an unnecessarily sharp quantization decision.

As shown in Table 5, the vast majority (93.19%) of the errors originally produced by the model are off-by-one errors. Among the errors successfully corrected by our method, the majority (88.9%) are also off-by-one cases, confirming that the dual-stream intervention primarily targets the boundary-crossing failure mode emphasized throughout the paper.

## J.3. Generalization Analysis

To assess the robustness of our approach across different architectures and task complexities, we extend our evaluation to Gemma-3-4B-IT (10-digit addition) and Qwen3-8B (12-digit addition), as detailed in Table 6.

In this broader analysis, we introduce two additional metrics to provide a holistic view of performance: **Question Accuracy (Ques Acc)** as explained above, and **Modified Rate**, which quantifies the frequency with which the inference-time method alters the original model output.

As shown in Table 6, the Dual-Stream Probe maintains competitive performance. We note that in some cases (particularly Qwen3-8B), the Dual-Stream Probe yields slightly lower accuracy metrics compared to the strict Replacement method. Crucially, these results serve as a causal validation of the IRST geometry rather than a mere performance boost. The ability to correct errors using only internal consistency (Dual-Stream) implies that the model retains correct latent information even when the output quantization fails. The Replacement method, while effective, relies on the probe's external supervision; the Dual-Stream method relies on the model's own internal coherence.

# K. Layer-Wise Analysis of Internal Activations

## K.1. Probe Analysis

To understand where and how the arithmetic logic emerges within the model , we conducted a comprehensive layer-wise probing analysis using Qwen3-4B. We use the same dataset and dataset split ratio as in Appendix J. We trained separate MLP probes on the hidden states of each layer (0–36) on all positions to predict six target variables which are consistent with the main text. The specific definitions of these targets are as follows:

**Correctness** A binary label indicating whether the model's generated digit successfully matches the ground truth.

**Carry Potential** A continuous regression target $\Phi_p$, described in Section 5.1.For evaluation and testing, the floor function is applied to interpret this continuous prediction as a discrete integer carry value.

**Ground Truth** The mathematically correct digit expected at the current position (integer label 0–9).

**Input Carry** The actual integer carry value transmitted from the previous column (integer label 0-2 in our specific task).

**Model Output** The specific digit token actually generated by the model (integer label 0–9).

**Raw Sum** The local sum of the operand digits in the current column modulo 10, excluding any incoming carry (integer label 0–9).

From the results in Figure 19, across all arithmetic-related variables (Raw Sum, Ground Truth, Model Output), we observe a distinct step-function behavior. The information regarding the specific digit values remains nearly imperceptible (accuracy close to random guessing) throughout the early and middle layers. A sudden informational surge occurs at Layer 24. The regression error for the continuous carry potential remains high in early layers but drops at Layer 24, coinciding with the emergence of the Raw Sum. The input carry information shows a more volatile trajectory in intermediate layers before stabilizing and peaking alongside the other metrics. This synchronization supports our Dual-Stream hypothesis: the model appears to disentangle and resolve both the local arithmetic facts (Raw Sum) and the global context (Carry) simultaneously to synthesize the final output.

To partially localize where the carry signal is written, we separately probe the outputs of the attention and FFN blocks. As shown in Figure 20, the attention modules exhibit the sharpest stepwise gains, whereas the FFN outputs mostly lag behind and mirror these updates. This pattern suggests that carry information is first consolidated by attention and then propagated through a staged feed-forward refinement process.

*Table 5.* **Error-type analysis** of the Dual-Stream Consistency method on Qwen3-4B for 10-digit addition at position $p = 4$.

| ERROR TYPE | FIXED | UNFIXED | TOTAL |
|---|---|---|---|
| OFF-BY-ONE ($\pm1$) | 96 | 561 | 657 |
| OTHER ERRORS | 12 | 36 | 48 |
| TOTAL | 108 | 597 | 705 |

*Table 6.* Performance comparison on Qwen3-4b, Gemma-3-4b-it and Qwen3-8b using various inference-time correction methods.

| METHOD | TOKEN ACC | QUES ACC | MODIFIED RATE | TP CORRECTION | FP PRESERVATION |
|---|---|---|---|---|---|
| *Model: Qwen3-4B    Task: 10-digit addition (3 terms)* | | | | | |
| ORIGINAL | 0.8626 | 0.2810 | / | / | / |
| STEERING | 0.8827 | 0.2820 | 0.0775 | 0.3058 | 0.9697 |
| REPLACEMENT | 0.8913 | 0.3000 | 0.0710 | 0.3173 | 0.9765 |
| RE-PROMPTING | 0.7990 | 0.2610 | 0.1040 | 0.0008 | 0.9998 |
| DUAL-STREAM PROBE ($\delta = 0$) | 0.8727 | 0.2320 | 0.1240 | 0.4439 | 0.9407 |
| DUAL-STREAM PROBE ($\delta = 0.05$) | 0.8887 | 0.2900 | 0.0844 | 0.3551 | 0.9675 |
| DUAL-STREAM PROBE ($\delta = 0.1$) | **0.8956** | **0.3240** | 0.0646 | 0.3046 | 0.9813 |
| *Model: Gemma-3-4B-IT    Task: 10-digit addition (3 terms)* | | | | | |
| ORIGINAL | 0.7286 | 0.3170 | / | / | / |
| STEERING | 0.8210 | 0.3010 | 0.1417 | 0.3319 | 0.9531 |
| REPLACEMENT | 0.8499 | **0.3780** | 0.1085 | 0.3405 | 0.9732 |
| RE-PROMPTING | 0.7141 | 0.3210 | 0.1065 | 0.1757 | 0.9133 |
| DUAL-STREAM PROBE ($\delta = 0$) | 0.8586 | 0.3470 | 0.1297 | 0.4046 | 0.9643 |
| DUAL-STREAM PROBE ($\delta = 0.05$) | **0.8598** | 0.3730 | 0.0967 | 0.3264 | 0.9837 |
| DUAL-STREAM PROBE ($\delta = 0.1$) | 0.8492 | 0.3580 | 0.0841 | 0.2726 | 0.9897 |
| *Model: Qwen3-8B    Task: 12-digit addition (3 terms)* | | | | | |
| ORIGINAL | 0.9329 | 0.5630 | / | / | / |
| STEERING | 0.9470 | 0.5770 | 0.0784 | 0.3108 | 0.9688 |
| REPLACEMENT | **0.9547** | **0.6440** | 0.0391 | 0.4222 | 0.9891 |
| RE-PROMPTING | 0.8806 | 0.5360 | 0.0738 | 0.0843 | 0.9379 |
| DUAL-STREAM PROBE ($\delta = 0$) | 0.9187 | 0.4390 | 0.0782 | 0.4439 | 0.9688 |
| DUAL-STREAM PROBE ($\delta = 0.05$) | 0.9378 | 0.5350 | 0.0425 | 0.3315 | 0.9838 |
| DUAL-STREAM PROBE ($\delta = 0.1$) | 0.9462 | 0.5710 | 0.0266 | 0.2638 | 0.9933 |

## K.2. UMAP Visualization

To further interpret the dynamic evolution of representations, we employ UMAP to perform dimensionality reduction on the hidden states of intermediate layers. Using the same Qwen3-4B model and the dataset described in Appendix J, we apply **Aligned UMAP** across all layers. Unlike standard UMAP, this technique enforces temporal consistency by regularizing the projection of each layer based on the geometry of the preceding layer. The results are visualized in Figure 21, and the complete evolutionary dynamics are provided as `GIF` files in the supplementary materials.

The results reveal that from Layers 1 to 23 (Figure 21a-21e), samples exhibit a persistent $10 \times 10$ hierarchical clustering structure, with ten macro-clusters representing the previously generated digit (Position 3), each contain ten micro-clusters. This structured organization remains stable throughout the early layers, with only minor shifts and local expansions or contractions. The precise mechanism of micro-clusters remains to be fully elucidated; we hypothesize they may represent the residual encoding of previously generated digits (e.g., the value at Position 2) or latent carry signals.

Consistent with the sudden informational surge observed in our probing experiments, a dramatic mutation occurs at Layer 24 (Figure 21f), where the hierarchical clusters collapse into an elliptical manifold. In the late layers (Figure 21g-21i), this manifold gradually dissociates into distinct IRST curves. This transition marks the point where the model resolves the latent variables in explicit arithmetic logic.

Notably, the IRSTs in Figure 21i appear disjoint, contrasting with the continuous manifold in Figure 2. This fragmentation is an artifact of Aligned UMAP: by penalizing large displacements to ensure temporal continuity, the distinct clustering of early layers acts as a geometric anchor that visually tears the emerging manifold. Thus, while Figure 2 depicts the intrinsic geometry, Figure 21i captures the path-dependent evolution.

## L. Probe vs. Logit Lens: Evidence of Latent Arithmetic States

To further validate that the geometric structures (IRSTs) captured by our probes represent intrinsic internal computations rather than trivial output reflections, we compared the performance of our Linear Probes against the **Logit Lens** technique

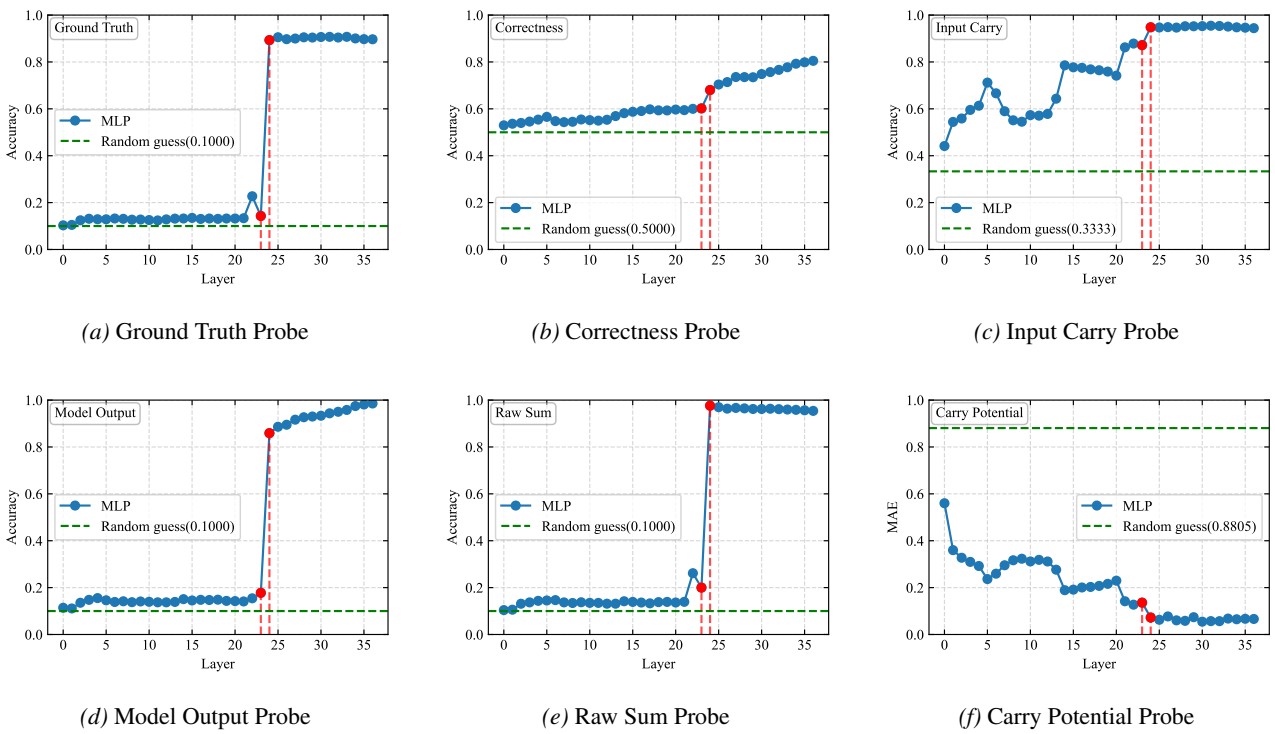

*(a)* Ground Truth Probe      *(b)* Correctness Probe      *(c)* Input Carry Probe

*(d)* Model Output Probe      *(e)* Raw Sum Probe      *(f)* Carry Potential Probe

*Figure 19.* Layer-wise performance of different probes trained on Qwen3-4B combing all positions.

(nostalgebraist, 2020).

The Logit Lens applies the final layer's unembedding matrix $W_U$ to intermediate hidden states $\boldsymbol{h}_p^{(l)}$ to project them directly into the vocabulary space. We evaluated both methods on identifying the Ground Truth Digit (GT) and the Model's Final Output Digit (Pred) across all 36 layers. The results for 3-term and 4-term addition are shown in Figure 22.

## L.1. Analysis of the Decoding Gap

The comparison reveals three distinct phases of processing:

1. **Preparation Phase (Layers 0–23):** Both probes and Logit Lens show near-random accuracy ($< 10\%$), suggesting that the specific digit information for the current position is not yet fully localized or linearly decodable.

2. **Latent Computation Phase (Layers 24–29):** This is the critical window where our geometric analysis focuses. Linear probes achieve high accuracy ($> 90\%$), indicating that the model has internally resolved the arithmetic state (Raw Sum and Carry). However, the Logit Lens accuracy remains low. This discrepancy proves that the internal representation at this stage is *orthogonal* or *unaligned* with the static word embedding space. The IRST structure exists here as an abstract mathematical representation, decoupled from the specific token IDs.

3. **Readout Phase (Layers 30–36):** The Logit Lens accuracy sharply rises to match the probes, indicating that the internal state is finally rotated into the vocabulary space for token generation.

This comparison strongly justifies the necessity of training probes (and analyzing the resulting geometry) rather than relying on the Logit Lens. The Logit Lens would miss the crucial "Latent Computation Phase" where the core arithmetic logic—and the geometric slippages we describe—actually unfold.

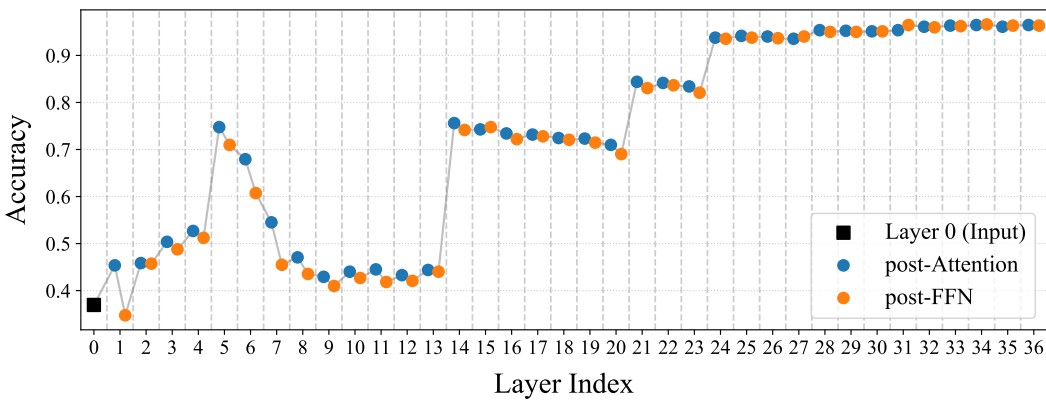

*Figure 20.* **Layer-wise input-carry decoding accuracy for attention and FFN outputs.** Attention blocks exhibit sharper stepwise gains, while FFN outputs largely follow these updates, suggesting that carry information is consolidated through a staged pipeline across layers.

*(a)* Layer 3          *(b)* Layer 6          *(c)* Layer 9

*(d)* Layer 15          *(e)* Layer 23          *(f)* Layer 24

*(g)* Layer 30          *(h)* Layer 33          *(i)* Layer 36

*Figure 21.* Layer-wise Alighed UMAP visualization.

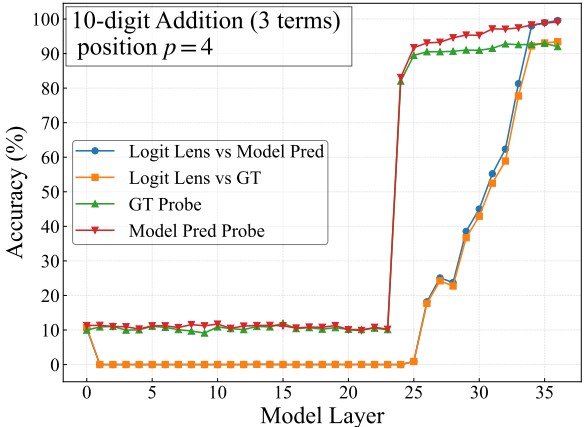 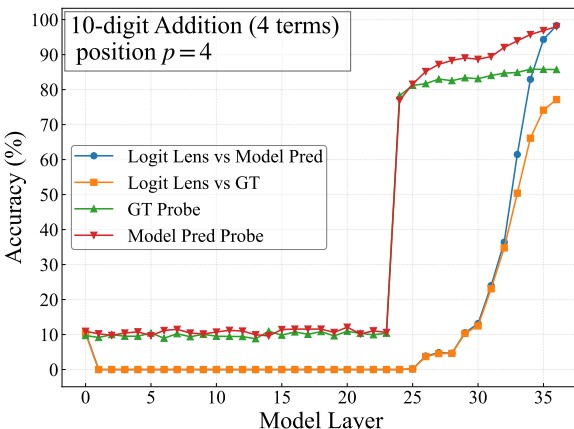

*Figure 22.* **Layer-wise Performance Comparison: Linear Probes vs. Logit Lens.** The Green and Red lines represent the accuracy of linear probes trained on $\boldsymbol{h}_p^{(l)}$ for the Ground Truth (GT) and Model Prediction (Pred), respectively. The Blue and Orange lines represent the accuracy of the Logit Lens (applying the unembedding matrix directly). **Key Observation:** There is a significant *decoding lag*. Probes successfully decode the arithmetic state (Accuracy $> 80\%$) starting around Layer 24, whereas the Logit Lens fails to extract meaningful information until Layer 30 (the "readout phase"). This gap indicates that the arithmetic computation occurs in a latent geometric subspace (the IRST) that is linearly separable but not yet aligned with the output vocabulary.

