# OpenReview forum: "The Shape of Addition: Geometric Structures of Arithmetic in Large Language Models"
_ICML.cc/2026/Conference — ICML 2026 regular_

### Official Review · Reviewer_Ywj3 · 2026-02-20

**Soundness:** 3
**Presentation:** 3
**Significance:** 3
**Originality:** 3
**Overall Recommendation:** 5
**Confidence:** 3

**Summary:**

This paper investigates why large language models remain fragile on multi digit, multi operand addition even when internal activations appear to contain information that supports the correct answer. The authors analyze residual stream activations during three term long addition and report a geometric organization of representations into digit basins with finer carry dependent structure. They introduce Iso Raw Sum Trajectories as continuous threads that connect states with the same raw sum while traversing adjacent digit basins as carry varies. Based on these observations, they propose a Noisy Quantization Model in which the model maintains a continuous carry potential and arithmetic errors arise when neural noise pushes this latent value across integer quantization thresholds, yielding predictable off by one failures. Finally, they propose an inference time dual stream consistency check using probes to detect and correct suspected quantization failures and report improvements under several settings.

**Compliance With Llm Reviewing Policy:**

Affirmed.

**Final Justification:**

The additional clarifications in the rebuttal addressed my main concerns

**Key Questions For Authors:**

- The experimental coverage seems fairly narrow, mostly focusing on Qwen3 and Gemma, with one Qwen3 8B setting. Could you evaluate additional model families with different tokenizers and training recipes, with other language models, to demonstrate that IRST and the carry threshold behavior are not specific to a single lineage.

- Since IRST is a central claim, I am still worried that the structure might be partly driven by UMAP choices. Could you provide a projection independent way to identify IRST and carry fibers in the original activation space, and also report stability across multiple UMAP hyperparameters and random seeds.

- The definition of the carry potential Phi is critical to the noisy quantization narrative, but it is not fully clear to me how Phi is computed. Please clarify whether Phi is derived from arithmetic ground truth, estimated by a probe from activations, or defined in another way, and include a sanity check that the main error versus Phi pattern remains under alternative Phi constructions or probe settings.

- The paper would be much more convincing if the threshold crossing mechanism were supported by a direct intervention rather than only correlational fits. Have you tried applying small controlled perturbations along the proposed carry direction near integer thresholds to see whether the output flips to adjacent digits as predicted, while the same perturbations far from thresholds have little effect.

-  Other questions and concerns please see the weakness section

**Limitations:**

No. The paper should more explicitly discuss limitations related to reliance on probes, sensitivity of UMAP based geometry. A brief discussion of potential misuse of inference time correction methods would also be appropriate.

**Strengths And Weaknesses:**

# Strengths

- The paper has a coherent end to end story that links representation analysis, probing, an explicit mechanistic hypothesis, and an inference time correction procedure.

- The core empirical pattern is plausible and aligns with known carry related failure modes in long addition. Errors concentrate near boundaries between adjacent digit regions and the proposed threshold crossing story is a reasonable explanation.

- The correction method is simple and testable. The tradeoff between true positive correction and false positive preservation is reported and the delta parameter gives a controllable knob.
- Understanding why models fail on simple deterministic tasks remains relevant for interpretability and reliability. If the geometric and quantization framing generalizes, it could influence follow up work on other discrete reasoning tasks.

- The paper is generally well written and the narrative is easy to follow from empirical observations to the proposed mechanism and the correction method.

# Weakness
- The strongest claims rely heavily on low dimensional UMAP visualizations. UMAP can distort neighborhood structure and can be sensitive to hyperparameters and random seeds. The paper would be more convincing with projection independent evidence that IRST and carry fibers exist as intrinsic structure in the original activation space.
-
- The core mechanism is still supported mainly by correlational evidence. The carry potential threshold story is plausible, but alternative explanations could also produce similar patterns, such as readout misalignment with the vocabulary basis, distributional artifacts near carry boundaries, or generic uncertainty near decision boundaries. Stronger causal intervention tests are needed to rule out these alternatives.

- Probe based evidence faces the standard concern that decodable does not imply used. Table 1 shows high probe accuracy for several variables, but this does not establish that the model relies on those directions for its output. This is especially important because carry potential is central to the proposed mechanism.

- The definition and estimation of carry potential needs to be fully clarified to avoid circularity. If carry potential is constructed in a way that already encodes threshold structure, then the bathtub shaped error rate curve may be less informative as a mechanism test.

- The correction evaluation needs clearer baseline fairness and stronger robustness reporting. Some baselines appear weak for token level correction, and results should be supported by multiple seeds and confidence intervals. It is also important to show whether the method mainly fixes off by one errors or changes error types.

---

> ### Author Rebuttal · Authors · 2026-03-30
>
> Sincerely thank you for your time and constructive feedback. We have carefully addressed your comments below.
> All newly added figures are prefixed with "R" (e.g., Fig. R1) for distinction, which can be viewed in **[this link](https://anonymous.4open.science/r/irst/README.md)**.
>
> **Q1: Projection-independent evidence & UMAP stability.**
> >To address concerns regarding UMAP dependence, we provide V-shaped ablations for all trajectories (Fig. R2). The correlation with the continuous carry potential $\Phi$ holds quantitatively across trajectories. As described in Sec. 9, $\mathcal{T}_8$ and $\mathcal{T}_9$ deviate because the 9-to-0 boundary lacks a continuous connection, intrinsically precluding slippage. Furthermore, extensive UMAP ablations across diverse hyperparameters and seeds (Fig. R1) confirm that the structures are stable.
>
> **Q2: Circularity concern: Definition of Carry Potential $\Phi$ and sanity check.**
> >We thank the reviewer for highlighting this. We clarify that there is no circularity. As explicitly defined in Section 5.1, Eq. (3), $\Phi_p$ is a strict, objective mathematical property derived solely from the ground-truth input digits. It is NOT estimated by any probe or derived from activations.
> >
> >To fulfill the sanity check request, we evaluated alternative $\Phi$ constructions by truncating the analytical look-ahead window to $K$ rightward digits: $\Phi_p^{(K)} = \sum_{j=1}^{\min(K, P-1-p)} \frac{r_{p+j}}{10^j}$. As shown in Fig. R11, while extreme truncation ($K=1$) slightly degrades the $R^2$ due to the loss of continuous resolution, a window of just $K=2$ perfectly recovers the periodic bathtub curve and high $R^2$. These results indicate that the observed threshold-crossing phenomenon is largely robust to how the potential is analytically defined.
>
> **Q3: Stronger causal intervention tests (Decodable vs Used).**
> >Appendix C already details a controlled causal steering experiment, where we actively intervened by perturbing the activation vectors strictly along the probed "carry potential" directions. The results demonstrate that modifying this specific geometric direction predictably alters the model's final output digit. For more native-space experiments, refer to Fig. R2.
>
> **Q4: Baseline fairness, multiple seeds, and confidence intervals.**
> >We agree that rigorous statistical validation is essential for evaluating inference-time intervention methods.
> >
> >To address this, we have re-evaluated all correction methods across 5 random seeds and calculated the 95% Confidence Intervals. The updated results for the Qwen3-4B model on the 10-digit addition task (3 terms) are presented below.
> >
> >| METHOD | TOKEN ACC (%) | TP CORRECTION (%) | FP PRESERVATION (%) |
> >| :--- | :---: | :---: | :---: |
> >| ORIGINAL | 86.4 ± 0.5 | / | / |
> >| STEERING | 88.2 ± 0.7 | 28.0 ± 4.9 | 97.4 ± 0.3 |
> >| REPLACEMENT | **89.6 ± 0.1** | 31.6 ± 2.1 | 98.0 ± 0.4 |
> >| RE-PROMPTING | 77.9 ± 1.0 | 9.7 ± 0.6 | 88.6 ± 0.8 |
> >| OURS ($\delta = 0$) | 87.4 ± 0.8 | **42.9 ± 0.9** | 94.5 ± 0.8 |
> >| OURS ($\delta = 0.1$) | 89.4 ± 0.4 | 28.9 ± 1.8 | **98.3 ± 0.5** |
> >
> >The marginal performance gap between Replacement and Dual-Stream Probe stems from mechanistic differences. Dual-Stream Probe deliberately cannot correct errors where the model's own latent raw_sum or carry representation is fundamentally corrupted. Furthermore, our method achieves higher TP CORRECTION and FP PRESERVATION rates.
>
> **Q5: Does the correction method mainly fix off-by-one errors?**
> >We analyzed the error types for the Qwen3-4B model on the 10-digit addition task (at position $p=4$). The error-shift matrix after applying our Dual-Stream Consistency method is presented below:
> >
> >| **Error Type**           | **Fixed**    | **Unfixed**  | **Total**    |
> >| ------------------------ | ------------ | ------------ | ------------ |
> >| **Off-by-One ($\pm 1$)** | 96| 561 | 657 |
> >| **Other Errors**         | 12 | 36  | 48 |
> >| **Total**                | 108| 597| 705 |
> >
> >As shown, the vast majority (93%) of the errors originally generated by the model were off-by-one errors. Among all the errors fixed by our method, the vast majority (89%) were also off-by-one errors.
>
> **Q6: Verification on more model families.**
> >Regarding broader model families, analyzing models like Llama is technically constrained because their lack of strict single-digit tokenization disrupts the position-wise geometric alignment required in our analysis. Thus, as an alternative, we evaluated a larger MoE model, Qwen3-30B-A3B (Fig. R12) and a toy model (Fig. R9, R10, see our response to Reviwer TsXj Q5). The distinct emergence of IRSTs confirms that this geometric structure can scale to both smaller toy models and larger MoE architectures.
>
> **Q7: Misuse risks in Limitations.**
> >We fully agree that a more targeted Limitations section is necessary. We will certainly expand the Limitations section in the revised version to discuss these open questions.

---

> > ### Author Rebuttal · Reviewer_Ywj3 · 2026-04-03
> >
> > Thank you for the detailed rebuttal. The additional clarifications and analyses address my main concerns. The broader model-family evidence is still somewhat limited, but I think the response is reasonable given the tokenization constraints you explained. I do not have further questions at this point. I've changed my recommendation to 5.

---

> > > ### Author Response · Authors · 2026-04-04
> > >
> > > Thank you again for the thoughtful feedback and the encouraging reassessment after our rebuttal. We are glad that the added analyses addressed your main concerns. We also appreciate your note on the current scope of the model-family evidence, and we will make this limitation clearer in the final version. Thank you again for your time and updated assessment.

---

### Official Review · Reviewer_TsXj · 2026-02-24

**Soundness:** 2
**Presentation:** 4
**Significance:** 2
**Originality:** 3
**Overall Recommendation:** 5
**Confidence:** 4

**Summary:**

This paper investigates the disconnect between the internal computations of LLMs and their frequent "off-by-one" errors in multi-digit arithmetic. By analyzing the residual stream activations of models like Qwen3 and Gemma-3, the authors identify the Iso-Raw-Sum Trajectory (IRST), a hierarchical topological manifold organized into macroscopic digit basins (representing categorical output) and microscopic carry fibers (representing internal carry states). The researchers propose the Noisy Quantization Model to explain arithmetic failures as Topological Slippages, where internal neural noise pushes a continuous, latent Carry Potential across discrete quantization thresholds, resulting in a predictable "bathtub" error distribution. Ultimately, the study validates this geometric framework through a dual-stream consistency check that successfully corrects outputs during inference by aligning local raw sum signals with global carry potentials, proving that models often retain correct mathematical information even when their discrete token selection is erroneous

**Compliance With Llm Reviewing Policy:**

Affirmed.

**Final Justification:**

Authors fullsome responses to my review questions led me to increase my overall recommendation from 4 to 5

**Key Questions For Authors:**

Q1: Do you think the Qwen model could reduce slippage with more training learning more accurate fibre/basin values. Alternatively could the fibers and basins be multi-used for related-but-different model features with possible slightly conflicting needs?

Q2: The https://arxiv.org/abs/2402.02619 paper provides 99.999% accurate 10-digit addition/subtraction toy models. Do you think the accuracy comes from fibre / basins being better defined? Or just by the single-task nature of the model?

**Limitations:**

Yes

**Strengths And Weaknesses:**

Strengths:
- Choice of models (Qwen3-4B and Gemma3) is good
- Choice of dataset (10K addition problems, each with 3 x 10-digit integers) is good.
- Great figures. Great visualization of errors as topological slippage - especially figures 2 and 5.
- I'm convinced you’ve detailed how the model represents addition in spectral space. As I see it, you’ve provided good evidence that you've traced the errors in tokens output back to inaccuracies in continuous values in the spectral space representation. This is a good shift that gets us closer to how the model does calculations.

Weaknesses:
- Definitions like "topological slippage" seem more like empirical descriptions of observed inaccuracy behaviors in spectral space and less as clearly defined mechanisms. This definition is useful but the paper reads as if this is a fundamental insight whereas I view it as useful metric from a new representation viewpoint. It is an empirical insight rather than a newly explained mechanism. I hope my comment is understandable.
- Quote "This spatial distribution suggests that arithmetic failures are essentially geometric instabilities, where the internal representation fails to settle into the correct topological fiber." This is true but more empirical observation than insightful. What does "settle into" mean? Are we just viewing continuous-value inaccuracies? Where do these inaccuracies come from? Possibly inaccuracries in params used in the calculations baked into the model during training?
- The paper would be further strengthened by a discussion of how the inaccuracies in continuous values in the spectral space representation came about, whether they can be "fixed" during training or at inference time.

Typos:
- Quote: “coexistence of these conflicting signals”. Why are they conflicting? Aren’t they coexisting / complementary?
- The terms fibers and basins are great but are not familiar to me. They were introduced very early and weren’t 100% clear to me until Figure 2 making it harder to read the paper end to end. Consider referencing Figure 2 earlier.
- Quote "However, due to the dilution of attention over long contexts and limited precision in superposition, this estimation is corrupted by neural noise". Make it clearer that you are positing this.
- Gemma3 is first mentioned in the Conclusion. Add a reference to it earlier in paper. (I see it is used in the Appendix repeatedly.)

---

> ### Author Rebuttal · Authors · 2026-03-30
>
> Sincerely thank you for your time and constructive feedback. We have carefully addressed your comments below.
> All newly added figures are prefixed with "R" (e.g., Fig. R1) for distinction, which can be viewed in **[this link](https://anonymous.4open.science/r/irst/README.md)**.
>
> **Q1: Mathematical definitions, wording & typos.**
> >We sincerely thank reviewers for the constructive feedback regarding the precision of our terminology. We agree that terms like "topological slippage" serve more as empirical descriptions of continuous-value inaccuracies rather than formal mathematical mechanisms. To avoid confusion, we will systematically tone down the formalism, such as replacing "topological slippage" with "geometric slippage".
> >
> >Regarding "conflicting," our intent was specifically to describe error states where a single vector encodes both the correct ground truth and an incorrect hallucinated output; we will clarify this and use "complementary" for the general case.
> >
> >Finally, while we already referenced Fig. 2 early in the Intro to visually ground UMAP, we agree the high density of novel terminology can still overwhelm readers. We will incorporate all your suggestions to improve readability, such as explicitly adding "we posit/hypothesize" for the noise mechanism and introducing Gemma3 earlier in the text.
>
> **Q2: What does "settle into" mean?**
> >We appreciate this constructive feedback. We agree that the arithmatic failures are essentially a continuous-value representational inaccuracy rather than a discrete failure to "settle into" a state. To reflect this precision, we will replace such wording with "align with the correct geometric trajectory" throughout the revision.
>
> **Q3: Source of inaccuracies and fixability during training/inference.**
> >Regarding the microscopic source, we hypothesize these inaccuracies arise from aggregate representational imprecision, including feature superposition, imperfect separation of related arithmetic states, and long-context mixing in the residual stream, rather than a single isolated cause.
> >
> >Regarding fixability, we believe such inaccuracies may be mitigated during training by improving the sharpness and separability of the relevant internal representations, though we did not directly test this here due to time limit. At inference time, our dual-stream intervention can be viewed as a selective correction method for some carry-boundary failures. But a complete solution for repairing the underlying representations requires further exploration in future work.
>
> **Q4: Could more training reduce slippage? Would features suffer from multi-use interference?**
> >Yes, we believe both of your intuitions are highly plausible and align with our geometric framework. First, more extensive training or task-specific fine-tuning would likely reduce slippage by learning better-separated fiber and basin values (See the next question Q5 for further reference). However, we suspect slippage might not disappear entirely in a general-purpose LLM.
> >
> >This naturally leads to your second point: multi-use interference (conceptually related to feature superposition). In general LLMs, shared representational directions are often forced to support multiple distinct semantic functions with slightly conflicting geometric requirements. This multi-use bottleneck inherently limits the arithmetic precision and contributes to the continuous neural noise. We will add both of these insightful points in our revised version.
>
> **Q5: Comparison with the 99.999% accurate toy model (arXiv:2402.02619).**
> >We thank the reviewer for this excellent connection. We hypothesize the answer is both: the single-task nature eliminates multi-use feature superposition, which consequently allows the model to form perfectly defined, isolated basins.
> >
> >To verify this, we evaluated two models from Quirke et al.(https://arxiv.org/abs/2402.02619). Their fully converged 10-digit model (quanta_add_d10) achieves 100% accuracy. Its UMAP (Fig. R9, right) shows perfectly isolated digit clusters with discrete carry sub-clusters, lacking the continuous connecting fibers that cause slippage.
> >
> >Interestingly, examining their under-converged 20-digit model (quanta_add_d20), we observe the similar continuous representations we described. Its errors are predominantly boundary-crossing off-by-one slippages (Fig. R9, left) that perfectly fit our NQM bathtub curve (Fig. R10). This suggests that geometric slippage is possibly a fundamental failure mode for continuous arithmetic representations, which single-task models overcome by learning perfectly isolated discrete basins.

---

> > ### Author Rebuttal · Reviewer_TsXj · 2026-04-02
> >
> > Thank you for your fullsome feedback. Much appreciated. I've changed my overall recommendation to 5.

---

> > > ### Author Response · Authors · 2026-04-04
> > >
> > > Thank you again for the thoughtful feedback and for your encouraging reassessment after reading our rebuttal. We will carefully incorporate the discussed revisions in the final version. If any further questions arise, we would be very happy to clarify them via the AC. We greatly appreciated.

---

### Official Review · Reviewer_gDbG · 2026-02-27

**Soundness:** 3
**Presentation:** 2
**Significance:** 3
**Originality:** 3
**Overall Recommendation:** 5
**Confidence:** 3

**Summary:**

The paper focuses on understanding why LLMs make simple addition mistakes. The authors found that the models treat math as a smooth internal signal, and small amounts of noise inside the model can disturb it, causing wrong answers. By monitoring how consistent these internal signals are, the authors proposed a framework to catch and correct the mistakes while they occur.

**Compliance With Llm Reviewing Policy:**

Affirmed.

**Final Justification:**

After considering the paper and the authors' rebuttal, I have raised my score to 5. My initial concerns focused on the clarity of presentation and the strength of the reported results. The authors addressed my questions, ran additional simulations that substantiated their claims, and promised to make the necessary changes.

**Key Questions For Authors:**

Overall, this is a nice piece of work that I am happy to support. However, I have the following questions for the authors:

- At what digit length or operand count does the model begin making errors? Do clean IRSTs without topological slippage exist for simpler additions, and how does the manifold degrade as complexity increases?

- Are the findings specific to addition? It would be interesting to see whether analogous structures emerge for other operations such as multiplication.

- Could the authors elaborate on which component, attention or feed-forward layers, is primarily responsible for encoding the carry signal?

- Why is Gaussian noise assumed in Equation 4?

- If time permits, training a small GPT-style model from scratch on this task would help determine whether the IRST geometry is a general transformer property.

**Limitations:**

The addition of a limitations section would strengthen the paper. The authors demonstrate that the model internally 'knows' the correct answer yet fails to output it. Whether this phenomenon is specific to arithmetic or instead reflects a broader issue in which continuous internal representations become lossy when collapsed to discrete tokens remains an important open question.

**Strengths And Weaknesses:**

- **Soundness:** The paper is highly technical and introduces several metrics, such as Continuous Carry Potential, the Iso-Raw-Sum trajectory as a geometric object, and dual-stream inference correction, to quantify and correct errors in the reasoning process for multi-digit addition.

- **Presentation:** I personally found the paper difficult to follow. In my view, it relies heavily on jargon without sufficient explanation, which makes it hard to understand for a general reader. For example, terms such as “bathtub error distribution,” “Iso-Raw-Sum trajectories,” “performance hierarchy,” “dual-stream consistency,” and “having a macroscopic backbone modulated by a microscopic texture” are introduced without clear, ordered explanation. These concepts may be meaningful, but they need to be defined and motivated more carefully. A reader should not need to read the entire paper in order to understand the terminology introduced on the first page. The paper would benefit from clearer explanations, better intuition-building, and a more accessible presentation. I recommend revising the manuscript to improve clarity and readability for a broader audience within the field.

- **Significance:**  The paper tackles an important problem and provides a novel geometric perspective on why LLMs struggle with multi-digit addition. By proposing a correction mechanism, it may open new avenues for improving the reliability of large language models.

- **Originality:** The model fill gap in the literature by itroducing a structured geometric model where the distance between carry-related elements affects the likelihood of errors through predictable structural shifts.

---

> ### Author Rebuttal · Authors · 2026-03-30
>
> Sincerely thank you for your time and constructive feedback. We have carefully addressed your comments below.
> All newly added figures are prefixed with "R" (e.g., Fig. R1) for distinction, which can be viewed in **[this link](https://anonymous.4open.science/r/irst/README.md)**.
>
> **Q1: Presentation and jargon in the Introduction.**
> >We agree that the Intro currently introduces several terms too quickly, which makes the paper harder to follow than necessary. In the revision, we will reorganize into a more linear progression, define each concept more explicitly at first use and add earlier intuition-building examples / figure references so that readers can understand the terminology without needing to read the entire paper first.
>
> **Q2: Relation between IRST/manifold degradation and problem complexity.**
> >To address how complexity affects the manifold, we analyzed a simpler 3-operand, 5-digit addition task (Fig. R6). When carry propagation is minimal (e.g., $p=4$), representations form tight, isolated digit basins without continuous connecting fibers, resulting in near-zero slippage. However, as carry variance increases ($p=3$), the continuous IRSTs explicitly emerge to encode this complexity. This confirms that clean, isolated clusters exist for simple arithmetic, and the continuous IRST (and its associated slippage) evolves directly from carry complexity.
>
> **Q3: Applicability to multiplication.**
> >As a preliminary test beyond addition, we visualized last-layer activations for 3‑digit × 3‑digit multiplication at $p=2$ (see Fig. R7). The global digit-basin organization remains visible, and most errors still lie between adjacent basins (off-by-one). However, the microscopic texture appears entangled in 2D. We hypothesize this reflects a higher-dimensional combinatorial grid necessary to encode multiple partial products and massive carry values. Due to time limit, we leave full analysis as future work.
>
> **Q4: Which component encodes the carry signal?**
> >To investigate whether Attention or FFN encodes the carry, we probed their respective outputs across layers (see Fig. R8, corresponding to Fig. 13). Attention modules are the primary drivers, exhibiting sharp accuracy jumps at specific layers (e.g., 5, 14, 21, 24), whereas FFNs predominantly lag and mirror these updates. The two-stage rise suggests a multi-phase computation. However, since decodable does not strictly imply used, causally localizing the precise arithmetic circuits via activation patching remains an important future direction.
>
> **Q5: Why is Gaussian noise assumed?**
> >We assume Gaussian noise as a minimal phenomenological approximation, not as an exact claim of the perturbation. The motivation is that the residual stream aggregates many contextual contributions, while superposition introduces numerous small interference terms; for such sums, Gaussian behavior is a natural first-order approximation under central-limit arguments, including settings with weak dependence.
>
> **Q6: Training a small GPT-style model to verify.**
> >We completely agree with this suggestion. Due to time limit, rather than training a new model from scratch, we analyzed the small transformers recently trained from scratch on arithmetic by Quirke et al. (https://arxiv.org/abs/2402.02619).
> >
> >Remarkably, their under-converged model (quanta_add_d20) exhibits the similar continuous IRST geometry and boundary-crossing slippages we observed in LLMs, also fitting our Noisy Quantization Model. We have provided detailed visualizations (Figs. R9 and R10) and a deeper discussion regarding this in our response to Reviewer TsXj Q5. This suggests that the IRST geometry is possibly a general representational intermediate state for transformers learning arithmetic.
>
> **Q7: Limitations section.**
> >We fully agree that a more targeted Limitations section is necessary. We will certainly expand the Limitations section in the revised version to explicitly discuss this open question.

---

> > ### Author Rebuttal · Reviewer_gDbG · 2026-04-02
> >
> > I thank the authors for addressing my questions. I am willing to raise my score provided that all the suggested changes are implemented in the paper.

---

> > > ### Author Response · Authors · 2026-04-04
> > >
> > > Thank you again for the constructive feedback and for the encouraging response to our rebuttal. We will make sure the discussed changes are fully incorporated in the revised paper, especially regarding clarity, claim calibration, and the added analyses. If any further questions remain, we would be very happy to clarify them via the AC.

---

### Official Review · Reviewer_6Gvt · 2026-03-12

**Soundness:** 3
**Presentation:** 3
**Significance:** 3
**Originality:** 3
**Overall Recommendation:** 5
**Confidence:** 3

**Summary:**

This paper studies the internal representations of Qwen3-4B and other models during addition of 10 digit integers. The main finding is that the final layer activations appear to organize into a Iso-Raw-Sum-Trajectory. They propose that errors arise from noisy quantization of a continuous carry potential Phi. Then they show that a dual-stream inference-time self-correction yields (modest) accuracy improvement.

**Compliance With Llm Reviewing Policy:**

Affirmed.

**Final Justification:**

Authors addressed concerns in the rebuttal. I maintain a positive assessment of the paper.

**Key Questions For Authors:**

1. Can you provide intrinsic dimensionality estimates (e.g., via nearest-neighbor dimension estimators) for the sets T_r in the native 2560-dimensional representation space? Without this, the "manifold" and "trajectory" characterizations lack empirical grounding.

2. The cosine-distance analysis (Figure 7, left) is shown only for T3. Does this V-shaped structure and smooth Φ-correlation hold quantitatively across all trajectories T0–T9? If some trajectories lack this structure, it would materially change the generality of the IRST framework.

3. What is the geometric signature of the ~10% of errors that violate the r̂_p ≈ r_p assumption? Do they cluster differently in activation space, or do they appear indistinguishable from carry-based errors? This would clarify the scope of the Noisy Quantization Model.
The dual-stream method (89.56%) barely outperforms the geometry-agnostic Replacement baseline (89.13%). What specific evidence supports that the IRST geometric structure provides explanatory or practical value beyond what a simple ground-truth probe already captures?

4. Can you show the full position-dependent error profile and bathtub curves, particularly at boundary positions (p=0, p=P-1)? The claim that p=4 is representative is undermined by your own observation of non-monotonic noise across positions (Appendix G).

**Limitations:**

yes

**Strengths And Weaknesses:**

### Strengths

The entire setup and story is compelling and culminates into a nontrivial contribution to interpretability.

The bathtub error curve (Fig 5) is a strong result. If errors come from noisy quantization of Phi, error rates should spike near integer boundaries and bottom out at half-integers. This holds with one free parameter (sigma approx 0.05) and replicates across three models. It's hard to explain this periodic structure under any errors-are-random hypothesis.

The paper is well structured, and the figures communicate effectively. Figures 2 and 3 convey the IRST clearly.

Generalization across different models strengthens the claim that this is a convergent computational strategy instead of being model-specific (one could always ask for more here and of course it would be interesting if certain models do not replicate these results, but I think the models analyzed here are sufficient for the paper).

The causal steering experiment in App C is sound.

### Weakness

UMAP dependence. The primary geometric claims depend on UMAP which is known to be highly sensitive to hyperparameters. The narrative is largely a property of the visualization and thus the chosen projection, not strictly a property of R^{2560}. The cosine distance analysis in App C is one such native-space measurement but is shown for only one trajectory. While I would not discount that UMAP is the right tool here, I think the narrative can more honestly reflect the situation. A clear indication of this is just how strikingly different Fig9 right is from the qwen models.

Overclaimed mathematical formalism. The paper overreaches on several mathematical terms. "topological slippage" and "topological manifold" do not seem to be referring to proper topological concepts at all and are more geometric. There are no intrinsic dimension estimates, no homology, etc. What the paper actually shows is that UMAP and tSNE projections display a certain clustered structure and that cosine distance in R2560 correlates with Phi with one trajectory. These notions are consistent with some notion of geometric structure, but the analysis in this paper does not make any of this precise. I'm assuming taht "topological" is referring to the consistent ordering of digit subclusters along the path which is not what is meant by "topological manifold" in mathematics.

The \hat{r}_p \approx r_p assumption is weaker than presented. The raw sum probe achieves 89% on a 28-class problem (App D). This is far above chance but means roughly 10% of errors are NOT carry-based slippages. The paper underplays this I think. The entire IRST framework and the Noisy Quantization Model assume carry errors are the dominant mechanism, but a 10% non-carry error rate seems to me to be quite significant?

---

> ### Author Rebuttal · Authors · 2026-03-30
>
> Sincerely thank you for your time and constructive feedback. We have carefully addressed your comments below.
> All newly added figures are prefixed with "R" (e.g., Fig. R1) for distinction, which can be viewed in **[this link](https://anonymous.4open.science/r/irst/README.md)**.
>
> **Q1: UMAP dependence, projection-independent evidence, and more V-shaped structures.**
> >Thank you for highlighting this. We agree that UMAP's global 2D arrangement is highly model- and projection-dependent. Our intended, narrower claim is that the local relational structure—digit clusters with carry-driven errors in adjacent transition regions—remains consistent across models. We will revise our wording to make this distinction explicit.
> >
> >For more ablation experiments, please see details in our response to reviewer Ywj3 Q1 (with Fig. R1, R2).
>
> **Q2: Overclaiming.**
> >We appreciate this critique. Please see our detailed revision in response to reviewer TsXj Q1.
>
> **Q3: Intrinsic dimensionality estimates.**
> >We estimated the intrinsic dimensionality (ID) of the IRSTs ($\mathcal{T}_0$,...,$\mathcal{T}_9$) directly in the native 2560-D residual space. See Fig. R3.
> >
> >Both linear and non-linear estimators confirm these representations occupy highly stable, exceptionally low-dimensional subspaces (Participation Ratio $\approx 5-6$; TWO-NN $\approx 15-17$; Levina-Bickel MLE$_{k=10} \approx 14-15$).
> >
> >Interestingly, pooling all trajectories yields a higher linear ID (PR $\approx 22.8$) but a lower non-linear ID (MLE $\approx 8.8$). A possible interpretation is that the full representation space relies on a shared macroscopic linear backbone, whereas individual trajectories carry richer, non-linear microscopic textures (see Fig. 2).
> >
> >We additionally examined layer-wise IDs for $\mathcal{T}_5$ and the pooled set (All) in Fig. R4. The two are broadly similar before layer 23. Between layers 23–31, $\mathcal{T}_5$ has slightly lower nonlinear ID (MLE) than All, while after layer 31 this trend reverses. One reading is that a relatively stable global scaffold forms first, followed by later refinement of trajectory-specific local structure, broadly consistent with App K.
> >
> >We will include these metrics in the paper as additional explaination.
>
> **Q4: The ~10% of errors that violate the raw sum assumption ($\hat{r}_p \approx r_p$).**
> >We clarify that the 89.35% in App D represents the probe's accuracy in recovering the latent 28-class raw sum, not the final token-level error rate. Thus, it does not strictly imply that ~10% of output errors are non-carry failures. However, we agree our previous wording overstated this. We will revise it accordingly.
> >
> >Additionally, we decoupled and visualized these error modes (Fig. R5). Unlike carry errors which systematically cluster at adjacent basin boundaries, the "raw wrong" non-carry errors are highly distinguishable. They scatter chaotically or collapse into unstructured spaces, confirming they represent distinct computational failures outside the scope of our continuous model.
>
> **Q5: Dual-stream method vs. Replacement baseline.**
> >While the Replacement baseline achieves high accuracy (89.13%), it operates as a black box. In contrast, the performance of the Dual-Stream method is significant because it empirically validates our geometric framework. It proves that the model internally disentangles arithmetic into distinct  raw-sum and carry signals.
> >
> >For further explanation, we conducted an ablation study ($\delta=0$, Qwen3-4B, 3-term addition) to decouple the geometric components:
> >
> >| Method | Token Acc (%) | Sample Acc (%) |
> >| :--- | :---: | :---: |
> >| Raw-Sum Probe + Carry Probe | 86.7 | 21.9 |
> >| Raw-Sum Probe + True Carry | 96.0 | 65.1 |
> >| Carry Probe + True Raw-Sum | 90.5 | 32.1 |
> >
> >As shown, Token Accuracy of Raw-Sum Probe + True Carry reaches 96.0%, which proves that the model's latent local computation is incredibly robust even when the output is wrong. Conversely, Token accuracy of Raw-Sum Probe + True Carry drops to 90.5%. This isolates the "noisy carry potential" as the bottleneck for arithmetic failures.
>
> **Q6: Full position-dependent error profile and bathtub curves.**
> >Thanks for this thorough check! We have plotted the error profiles across all positions (Fig. R13). Crucially, intermediate positions ($p=1 \dots 8$) consistently replicate the periodic bathtub curve, confirming $p=4$ can represent the steady-state. We clarify that our previous mention of "non-monotonic" in App G was based on preliminary rough tests; however Fig. R13  suggests that variance across intermediate positions is merely minor statistical noise.
> >
> >Nevertheless, boundary positions predictably deviate. We hypothesize that $p=0$ lacks sufficient autoregressive noise to trigger slippage, and $p=9$ has strict discrete potentials lacking continuous resolution. We will update App G with these complete profiles.

---

> > ### Author Rebuttal · Reviewer_6Gvt · 2026-04-03
> >
> > Thank you for the reply. This resolves any outstanding questions I had. I will maintain my positive score.

---

> > > ### Author Response · Authors · 2026-04-04
> > >
> > > Thank you again for the thoughtful feedback and for your positive assessment. We will carefully incorporate the discussed revisions in the final version. If any further questions arise, we would be very happy to clarify them via the AC.

---

### Decision · Program_Chairs · 2026-04-30

**Decision:**

Accept (regular)

**Comment:**

All reviewers agree this work shines new light on a widely observed failure of Transformers and offers a partial solution. Remaining concerns have been addressed by clarifications in the rebuttal and additional experimental results.